# MissNODAG: Differentiable Learning of Cyclic Causal Graphs from Incomplete Data

**Muralikrishnna G. Sethuraman**                                    *muralikgs@gatech.edu*
*School of Electrical and Computer Engineering*
*Georgia Institute of Technology*

**Razieh Nabi**                                                    *razieh.nabi@emory.edu*
*Department of Biostatistics and Bioinformatics*
*Emory University*

**Faramarz Fekri**                                          *faramarz.fekri@ece.gatech.edu*
*School of Electrical and Computer Engineering*
*Georgia Institute of Technology*

**Reviewed on OpenReview:** *https://openreview.net/forum?id=nNZXQ3QOGP*

## Abstract

Causal discovery in real-world systems, such as biological networks, is often complicated by feedback loops and incomplete data. Standard algorithms, which assume acyclic structures or fully observed data, struggle with these challenges. To address this gap, we propose MissNODAG, a differentiable framework for learning both the underlying cyclic causal graph and the missingness mechanism from partially observed data, including data *missing not at random*. Our framework integrates an additive noise model with an expectation-maximization procedure, alternating between imputing missing values and optimizing the observed data likelihood, to uncover both the cyclic structures and the missingness mechanism. We establish consistency guarantees under exact maximization of the score function in the large sample setting. Finally, we demonstrate the effectiveness of MissNODAG through synthetic experiments and an application to real-world gene perturbation data.

## 1 Introduction

Causal discovery, the process of identifying causal relationships from data, is fundamental across scientific domains such as biology, economics, and medicine (Spirtes et al., 2000; Sachs et al., 2005; Zhang et al., 2013; Segal et al., 2005; Imbens & Rubin, 2015). Understanding these relationships is crucial for predicting how systems respond to interventions, enabling informed decision-making in complex systems (Solus et al., 2017; Sulik et al., 2017; Sethuraman et al., 2023). Traditionally, causal relationships are modeled using *directed graphs*, where nodes represent variables, and directed edges encode cause-effect relationships.

Existing causal discovery methods are typically divided into two main categories: *constraint-based* and *score-based* approaches. Constraint-based methods, such as the PC algorithm (Spirtes et al., 2000; Triantafillou & Tsamardinos, 2015; Heinze-Deml et al., 2018), infer the causal structure by enforcing conditional independencies observed in the data, though they often struggle with scalability due to the large number of required conditional independence tests. Score-based methods, such as the GES algorithm (Meek, 1997; Hauser & Bühlmann, 2012), optimize a penalized score function, like the Bayesian Information Criterion, over the space of candidate graphs, usually employing greedy search techniques. There also exist *hybrid* methods which combine elements of both approaches, leveraging conditional independence tests alongside score optimization (Tsamardinos et al., 2006; Solus et al., 2017; Wang et al., 2017). Recent advances have introduced *differentiable* discovery methods, such as the NOTEARS algorithm (Zheng et al., 2018), which frames learning of a Directed Acyclic Graph (DAG) as a continuous optimization problem, enabling scalable and efficient solutions via gradient-based

methods. Following NOTEARS, several extensions have been developed for learning DAGs under various assumptions in observational settings (Yu et al., 2019; Ng et al., 2020; 2022; Zheng et al., 2020; Lee et al., 2019).

Most causal discovery methods make one or both of the following assumptions: (i) the data is fully observed, and (ii) the underlying graph is acyclic. However, real-world systems often violate these assumptions. Biological systems, such as gene regulatory networks, and socio-economic processes frequently exhibit feedback loops (cycles) (Sachs et al., 2005; Freimer et al., 2022), while missing data is common in practical applications (Getzen et al., 2023). These complexities significantly limit the applicability of standard causal discovery methods.

Missing data mechanisms are classified into three categories: Missing Completely At Random (MCAR), Missing At Random (MAR), and Missing Not At Random (MNAR) (Little & Rubin, 2019). One common approach to dealing with missing data involves discarding incomplete samples or excluding variables with missing data (Carter, 2006; Van den Broeck et al., 2015; Strobl et al., 2018), which might be suitable only in restrictive settings such as MCAR mechanisms while missingness rate is negligible. Otherwise, it leads to performance degradation as the missingness increases. Another common approach is imputation-based methods where the missing data is first imputed before applying causal discovery algorithm on the data. Some notable imputation algorithms include multivariate imputation by chained equations (MICE) (White et al., 2011), MissForest (Stekhoven & Bühlmann, 2012), optimal transport (Muzellec et al., 2020), and a few deep learning based approaches (Li et al., 2019; Luo et al., 2018). However, imputation-based methods typically assume that data are MAR, which can lead to bias when the data are actually MNAR, a common occurrence in practice (Singh, 1997; Wang et al., 2020; Kyono et al., 2021; Gao et al., 2022). Gain & Shpitser (2018) addresses MNAR data by using reweighted observed cases as input to the PC algorithm alongside a weighted correlation matrix. Additionally, Tu et al. (2019) extends the PC algorithm by incorporating corrections to account for both MAR and certain cases of MNAR, while also learning the underlying missingness mechanisms.

Furthermore, while the acyclicity assumption simplifies computations by factorizing joint distributions into conditional densities, many real-world systems feature cyclic relationships (Sachs et al., 2005; Freimer et al., 2022). Several approaches have been developed to relax the acyclicity assumption, allowing for cyclic causal graphs. For example, early work by Richardson (1996) extended the constrained-based framework to account for cycles, and Lacerda et al. (2008) provide an Independent Componenet Analysis (ICA) based causal discovery for linear non-Gaussian cyclic graphs. More recent score-based methods for learning cyclic graphs include (Huetter & Rigollet, 2020; Améndola et al., 2020; Mooij & Heskes, 2013; Drton et al., 2019). Additionally, methods such as those proposed by Hyttinen et al. (2012) and Huetter & Rigollet (2020) focus on learning cyclic graphs from interventional data. Sethuraman et al. (2023) further extended this line of approach to nonlinear cyclic directed graphs, eliminating the need for augmented Lagrangian-based solvers by directly modeling the data likelihood.

**Contributions.** In this work, we address two major limitations in causal discovery: the inability to handle informative MNAR data and the restriction to acyclic structures. We propose *MissNODAG*, a general framework that extends expectation–maximization (EM)–based causal discovery to cyclic graphs and arbitrary graphically represented MNAR mechanisms. MissNODAG alternates between imputing missing values and maximizing the expected log-likelihood of the observed data, unifying and extending prior approaches such as MissDAG (Gao et al., 2022). Following Sethuraman et al. (2023) and Behrmann et al. (2019), we employ residual normalizing flows to flexibly model data likelihoods in both linear and nonlinear structural equation models. Our framework accommodates MCAR, MAR, and MNAR missingness and can represent any MNAR process that admits a graphical specification. Through synthetic experiments, we show that MissNODAG outperforms state-of-the-art imputation techniques combined with causal learning on partially missing interventional data. Furthermore, we also provide consistency guarantees for exact maximization of the score function.

The paper is organized as follows. In section 2, we describe the problem setup and outline the modeling assumptions. Section 3 introduces the proposed expectation-maximization-based MissNODAG framework. In section 4, we validate MissNODAG on various synthetic datasets. Section 5 concludes the paper. All proofs are deferred to the appendix.

## 2 Problem Setup

A list of notations is provided in Appendix A for the ease of reference.

**Structural Causal Model.** Let $\mathcal{G} = (X, \mathcal{E})$ denote a possibly cyclic causal graph with a set of vertices $X = (X_1, \ldots, X_K)$, representing a vector of $K$ random variables connected by directed edges $\mathcal{E} \subseteq X \times X$. We assume the following *structural causal model* (SCM), also known as structural equation model (SEM), with additive noise terms to capture the functional relationships between variables in $\mathcal{G}$ (Bollen, 1989; Pearl, 2009):

$$X_k = f_k\big(\text{pa}_{\mathcal{G}}(X_k)\big) + \epsilon_k, \quad k = 1, \ldots, K, \tag{1}$$

where $\text{pa}_{\mathcal{G}}(X_k) = \{X_\ell \in X \mid X_\ell \to X_k \in \mathcal{E}\}$ denotes the parents of $X_k$ in $\mathcal{G}$. The function $f_k$ describes the relationship between $X_k$ and its parents, with $\epsilon_k$ as the exogenous noise term, assumed to be mutually independent (no unmeasured confounders), and collected as $\epsilon = (\epsilon_1, \ldots, \epsilon_K)$. We assume that self-loops (edges of the form $X_k \to X_k$) are absent in $\mathcal{G}(X)$, as this could lead to model identifiability issues (Hyttinen et al., 2012).

Let $\text{F}(X)$ collect the functions $f_k(\text{pa}_{\mathcal{G}}(X_k))$, for all $k$. The structural equations in eq. (1) can be written as follows:

$$X = \text{F}(X) + \epsilon. \tag{2}$$

Let id denote the identity map, so the *forward map* $(\text{id} - \text{F})$ maps $X$ to $\epsilon$. We assume that this mapping is bijective, ensuring the existence of $(\text{id} - \text{F})^{-1}$ (called the *reverse map*), and that both $(\text{id} - \text{F})$ and $(\text{id} - \text{F})^{-1}$ are differentiable. The former ensures that there is a unique $X$ for a given $\epsilon$, thus, we can express $X$ as $X = (\text{id} - \text{F})^{-1}(\epsilon)$. This assumption is needed for our developments in section 3.3, and is naturally satisfied when the underlying graph is acyclic (Mooij & Heskes, 2013; Sethuraman et al., 2023).

Intervention operations can be readily incorporated into eq. (2). In this work, we focus exclusively on surgical, or hard, interventions (Spirtes et al., 2000; Pearl, 2009). Graphically, a hard intervention corresponds to removing all incoming edges to the intervened variable. Following similar notational convention in (Hyttinen et al., 2012; Sethuraman et al., 2023), given a set of interventional targets $X_{\mathcal{I}}$, we can decompose $X$ into disjoint sets, $X = X_{\mathcal{I}} \cup X_{\mathcal{O}}$, where $X_{\mathcal{I}}$ represents the set of intervened variables in an interventional experiment, and $X_{\mathcal{O}}$ represents the set of purely observed variables. We denote the graph obtained after performing the interventions on $X_{\mathcal{I}}$ as $\text{do}(X_{\mathcal{I}})(\mathcal{G})$. Let $\mathbf{D} \in \{0, 1\}^{K \times K}$ be a diagonal matrix where $D_{kk} = 1$ if $X_k \in X_{\mathcal{O}}$, and 0 otherwise. Under this setting, the SEM in eq. (2) is now modified to:

$$X = \mathbf{D}\text{F}(X) + \mathbf{D}\epsilon + C, \tag{3}$$

where $C$ denotes a vector of size $K$ representing intervention assignments for variables in $X$. Specifically, $C_k = X_k$ if $X_k \in X_{\mathcal{I}}$, and $C_k = 0$ otherwise. Let $\epsilon_{\mathcal{O}}$ denote the exogenous noise terms corresponding to variables in $X_{\mathcal{O}}$. Let $p_{\epsilon_{\mathcal{O}}}(\epsilon_{\mathcal{O}})$ and $p_{X_{\mathcal{I}}}(X_{\mathcal{I}})$ be the joint probability densities of $\epsilon_{\mathcal{O}}$ and $X_{\mathcal{I}}$, respectively. We thus have,

$$p_X(X) = p_{X_{\mathcal{I}}}(X_{\mathcal{I}}) \, p_{\epsilon_{\mathcal{O}}}(\epsilon_{\mathcal{O}}) \, \big|\det J_{(\text{id}-\mathbf{D}\text{F})}(X)\big|, \tag{4}$$

where $\det J_{(\text{id}-\mathbf{D}\text{F})}(X)$ denotes the determinant of the vector-valued Jacobian matrix of the function $(\text{id}-\mathbf{D}\text{F})$ at $X$. See a proof in appendix B.1.

**Missing Data Model.** Given sampled data on $X$, let $R = (R_1, \ldots, R_K)$ be the vector of binary missingness indicators with $R_k = 1$ if $X_k$ is observed and $R_k = 0$ if $X_k$ is missing. We only observe a coarsened version of $X_k$, denoted by $Y_k$, which is deterministically defined as $Y_k = X_k$ when $R_k = 1$, and $Y_k = ?$ if $R_k = 0$. Let $Y = (Y_1, \ldots, Y_K)$ denote the coarsened variables. Additionally, we have access to $S = (S_1, \ldots, S_K)$ where $S_k$ is a binary indicator of intervention, such that $S_k = 0$ if $X_k$ is intervened on (i.e., $X_k \in X_{\mathcal{I}}$), and $S_k = 1$ otherwise. We assume we have $n$ i.i.d. copies of $(Y, R, S)$, and the dataset is denoted by $\mathcal{D} = \{y^{(i)}, r^{(i)}, s^{(i)}\}_{i=1}^n$, where $y^{(i)}, r^{(i)}, s^{(i)}$ represent the $i$-th observed values of $Y, R, S$.

We define a missing data model as the collection of distributions over the variables $X, R, Y$. By chain rule of probabilities, we can express $p(X, R, Y)$ as $p(X) \, p(R|X) \, p(Y|X, R)$. We refer to $p(X)$ as the *target law*, $p(R|X)$ as the *missingness mechanism*, $p(X, R)$ as the *full law*, while $p(Y|X, R)$ is the *coarsening mechanism*, which is deterministically defined. Borrowing ideas from the graphical models of missing data (Mohan et al., 2013; Nabi et al., 2025), we use graphs to encode assumptions about $p(X, R, Y)$.

Specifically, we assume that the relations between variables in the target law $p(X)$ are directed and can include cycles, and the missingness mechanism $p(R|X)$ factorizes according to a DAG, where $\mathrm{pa}_{\mathcal{G}}(R_k)$ can only be a subset of $X$ and $R \setminus R_k$. Finally, due to deterministic relations, $Y_k$ has only two parents $R_k$ and $X_k$. We denote these graphs by $\mathcal{G}_m = (V, \mathcal{E}_m)$, where $V = (X, R, Y)$, and $\mathcal{E}_m$ denotes the set of directed edges in the missing data graph. Two examples of missing data graphs (or $m$-graphs), with $K = 3$ substantive variables, are provided in figure 1; deterministic relations are drawn in gray.

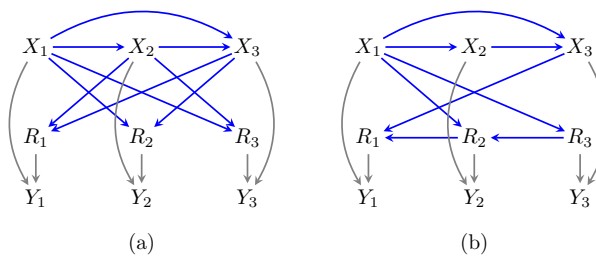

Figure 1: Example $m$-graphs with three variables illustrating: (a) An example of an MNAR mechanism where no edge of the form $R_i \rightarrow R_j$ exists; (b) An MNAR mechanism where $R$'s are connected and the full law is identifiable (no self-censoring and no colluder structures).

Graphically, an MCAR mechanism has no incoming edges into any missingness indicator in $R$, a MAR mechanism has parents of missingness indicators that are fully observed, and an MNAR mechanism involves missingness indicators with parents in $X$. Identifying the full or target law in an $m$-graph with MNAR mechanisms from observational data is not always possible. Previous work has extensively studied identification in graphical models of missing data (Bhattacharya et al., 2020; Nabi et al., 2020; Mohan & Pearl, 2021; Guo et al., 2023; Nabi et al., 2025; Guo & Nabi, 2026). Nabi et al. (2020) have shown that the full law in an $m$-graph is identified *if and only if* there are no edges of the form $X_k \rightarrow R_k$ (no *self-censoring*) and $X_j \rightarrow R_k \leftarrow R_j$ (no *colluders*). Therefore, we restrict the $m$-graphs considered in this work to be identifiable (formalized in assumption 1).

**Assumption 1.** *The missing data graphs under consideration have no edges of the form $X_k \rightarrow R_k$ (no* self-censoring*) and $X_j \rightarrow R_k \leftarrow R_j$ (no* colluders*).*

Given partially observed data from a set of interventional experiments, our objective is to learn the underlying full law that generated the sample. Specifically, this involves learning both the underlying target law and the missingness mechanism.

## 3 The MissNODAG Framework

We assume the target law $p(X)$ and the missingness mechanism $p(R|X)$ are parameterized by finite vectors $\theta$ and $\phi$, respectively. Thus, we write the full law $p(X, R)$ as $p(X, R|\theta, \phi) = p(X|\theta)\, p(R|X, \phi)$. In order to learn the full law, we proceed by maximizing the log-likelihood of the observed data law.

Let $\Gamma_i = \{k : r_k^{(i)} = 1\}$ and $\Omega_i = \{k : r_k^{(i)} = 0\}$ represent the sets of indices for the variables that are observed and missing, respectively, in the $i$-th sample; thus $y^{(i)} = x_{\Gamma_i}^{(i)}$. Consequently, the observed data law $p\big(x_{\Gamma_i}^{(i)}, r^{(i)}\big)$ can be written down as

$$p\big(x_{\Gamma_i}^{(i)}, r^{(i)} \mid \theta, \phi\big) = \int p\big(x_{\Gamma_i}^{(i)}, x_{\Omega_i}^{(i)}, r^{(i)} \mid \theta, \phi\big)\; dx_{\Omega_i}^{(i)}. \tag{5}$$

This integration is generally intractable due to marginalization over missing variables and the lack of a closed-form solution. We address this intractability in maximizing the observed data likelihood when data is generated from an $m$-graph, as described in section 2. First, we describe the score function to be maximized and establish the consistency result in section 3.1. Then, we provide an overview of the MissNODAG framework in section 3.2, followed by details on computing the log-likelihood for missing and observed variables, and discuss imputing the missing variables under linear and nonlinear SEMs in section 3.3.

### 3.1 Score Function

Assuming the data $\mathcal{D} = \{y^{(i)}, r^{(i)}, s^{(i)}\}_{i=1}^n$ is generated from an experiment where the full law $p(X, R)$ is represented via $m$-graphs, our goal is to learn the entire $m$-graph structure by maximizing, $\tilde{\mathcal{L}}(\mathcal{D}, \theta, \phi)$, a

regularized log-likelihood of the observed data law:

$$\max_{\theta, \phi} \sum_{i=1}^{n} \log p\big(x_{\Gamma_i}^{(i)}, r^{(i)} | \theta, \phi\big) - \lambda_1 \mathcal{R}(\theta) - \lambda_2 \mathcal{R}(\phi) \quad \text{s.t.} \quad h_1(\phi) = 0 \quad \text{and} \quad h_2(\phi) = 0, \tag{6}$$

where $\mathcal{R}(\cdot)$ is a regularization function that promotes sparsity, $\lambda_1, \lambda_2$ are regularization parameters, and the functions $h_1(\phi)$ and $h_2(\phi)$ are functions that restrict the missingness mechanism to the class of identifiable MNAR mechanisms, see section 3.3.3 for more details.

Given a family of $M$ interventional experiments with targets $\{X_{\mathcal{I}_m}\}_{m=1}^{M}$, under the population (infinite data) setting, the regularized observed data likelihood score for an m-graph $\mathcal{G}_m$, denoted as $\mathcal{S}(\mathcal{G}_m)$ is given by

$$\mathcal{S}(\mathcal{G}_m) := \sup_{\theta_{\mathcal{G}_m}, \phi_{\mathcal{G}_m}} \sum_{m=1}^{M} \mathbb{E}_{(X_\Gamma, R) \sim p_{\text{obs}}^{(m)}} \log p(X_\Gamma, R) - \lambda |\mathcal{G}_m|, \tag{7}$$

where $p_{\text{obs}}^{(m)}$ is the true data generating distribution for the $m$-th interventional setting, $|\mathcal{G}_m|$ combines $\mathcal{R}(\theta)$ and $\mathcal{R}(\phi)$ and denotes sparsity regularization on the m-graph (number of edges), and $\lambda$ denotes the combined sparsity regularization parameter. The following theorem establishes that, under standard assumptions assumptions, the exact maximization of $\mathcal{S}(\mathcal{G})$ results in a graph $\hat{\mathcal{G}}_m$ that is in the same Markov equivalence class as the ground truth graph $\mathcal{G}_m^*$. Formally,

**Theorem 2.** *Let $\{X_{\mathcal{I}_m}\}_{m=0}^{M}$ be a family of interventional targets, let $\mathcal{G}_m^*$ denote the ground truth directed mixed graph, $p^{(m)}$ denote the data generating distribution for $X_{\mathcal{I}_m}$, and $\hat{\mathcal{G}}_m := \arg\max_{\mathcal{G}_m \in \mathcal{G}_{id}} \mathcal{S}(\mathcal{G}_m)$, where $\mathcal{G}_{id}$ denotes the set of all m-graph that obey assumption 1. Then, under assumptions 1, B.7 to B.9 and B.11, and for a suitably chosen $\lambda > 0$, $\hat{\mathcal{G}}_m$ is $\mathcal{I}$-Markov equivalent to $\mathcal{G}_m^*$, denoted as $\hat{\mathcal{G}}_m \equiv_{\mathcal{I}} \mathcal{G}_m^*$.*

The proof for theorem 2 is provided in appendix B.3. This result extends the consistency guarantee established by Sethuraman & Fekri (2025) under complete data to the missing data setting. While exact maximization of the score function theoretically guarantees recovery of the ground truth Markov equivalence class, it is computationally infeasible in practice. Consequently, we resort to an Expectation-Maximization (EM) algorithm to optimize the score function (detailed in section 3.2). Although EM does not strictly maximize the observed data likelihood—limiting convergence to a stationary point—our experiments indicate that MissNODAG is capable of recovering the true target law graph when provided with all single-node interventions (see section 4).

## 3.2 The Overall Procedure

As discussed, computing $p(x_{\Gamma_i}^{(i)}, r^{(i)} | \theta, \phi)$ is generally intractable, with no closed-form solution. However, eq. (6) can be solved using the iterative penalized expectation-maximization (EM) method (Chen et al., 2014). Unlike imputation methods that directly sample missing values, the EM algorithm starts with an initial parameter $\Theta^0 = (\theta^0, \phi^0)$ and alternates between the following two steps at iteration $t$ until convergence:

**E-step:** Use the current estimates of the model parameters, $\Theta^t = (\theta^t, \phi^t)$, and the non-missing data to compute the expected log-likelihood of the full data, denoted by $\mathcal{Q}(\Theta|\Theta^t)$, and given by:

$$\mathcal{Q}(\Theta \mid \Theta^t) = \sum_{i=1}^{n} \mathbb{E}_{x_{\Omega_i}^{(i)} \sim p(\cdot | x_{\Gamma_i}^{(i)}, r^{(i)}, \Theta^t)} \Big[ \log p\big(x_{\Gamma_i}^{(i)}, x_{\Omega_i}^{(i)}, r^{(i)} \mid \Theta\big) \Big]. \tag{8}$$

**M-step:** Maximize $\mathcal{Q}(\Theta|\Theta^t)$, computed in the E-step, with respect to $\Theta$:

$$\Theta^{t+1} = \arg\max_{\Theta} \ \mathcal{Q}(\Theta \mid \Theta^t) - \lambda_1 \mathcal{R}(\theta) - \lambda_2 \mathcal{R}(\phi) \quad \text{s.t.} \quad h_1(\phi) = 0 \quad \text{and} \quad h_2(\phi) = 0. \tag{9}$$

We use stochastic gradient-based solvers to solve the maximization problem, alternating between updating the parameters of the target law, $\theta$, and the parameters of the missingness mechanism, $\phi$. Note that,

$$\sum_{i=1}^{n} \log p(x_{\Gamma_i}^{(i)}, r^{(i)} \mid \Theta) \geq \mathcal{Q}(\Theta \mid \Theta^t) - \text{const.} \tag{10}$$

This inequality indicates that we maximize a lower bound of the log-likelihood as we update the parameters during the M-step. More details on the derivation of eq. (10) and on the convergence analysis of MissNODAG are provided in appendix B.2.

### 3.3 Computational Details of the E-step

Computing the expected log-likelihood in the E-step can be challenging for a directed (cyclic) graph. First, let's look at $\log p\big(x_{\Gamma_i}^{(i)}, x_{\Omega_i}^{(i)}, r^{(i)}|\Theta\big)$ in eq. (8), which equals:

$$\log p_X\big(x_{\Gamma_i}^{(i)}, x_{\Omega_i}^{(i)}|\theta\big) + \log p\big(r^{(i)}|x_{\Omega_i}^{(i)}, x_{\Gamma_i}^{(i)}, \phi\big). \tag{11}$$

#### 3.3.1 Target Law

As per eq. (4), computing $\log p_X(X|\theta)$ in eq. (11) requires:

$$\log \big| \det J_{(\mathrm{id}-\mathbf{DF})}(X)\big|. \tag{12}$$

In the worst case, the Jacobian matrix may require gradient calls in the order of $K^2$. To that end, following Sethuraman et al. (2023), we employ *contractive residual flows* (Behrmann et al., 2019; Chen et al., 2019) to compute the log-determinant of the Jacobian in a tractable manner.

**Modeling the SEM in eq. (2).** We assume the SEM functions in $\mathrm{F}(X)$ in eq. (2) are Lipschitz with Lipschitz constant less than one. Such functions are called *contractive functions*. It then follows from *Banach fixed point* theorem (Rudin, 1953) that the mapping function $(\mathrm{id}-\mathbf{DF})$ is contractive and invertible.

We use neural networks to model the contractive functions in $\mathrm{F}(X)$. As shown by Behrmann et al. (2019), neural networks can be constrained to be contractive during the training phase by rescaling the layer weights by their corresponding spectral norm. While contractivity is a sufficient condition for the existence of $(\mathrm{id}-\mathrm{F})^{-1}$, it is not necessary. When the underlying graph governing the target law $p(X)$ is a DAG, it is possible to have non-contractive functions in $\mathrm{F}(X)$ for which $(\mathrm{id}-\mathrm{F})^{-1}$ exists; see (Sethuraman et al., 2023) for more details.

Naive implementation of a neural network may not produce promising results as it may introduce self-cycles. To circumvent this issue and simultaneously add sparsity penalization, we introduce a *dependency mask* matrix $\boldsymbol{M} \sim \{0,1\}^{K \times K}$ to explicitly encode the dependencies between the nodes in the graph, with zero diagonal entries to mask out the self-loops. Thus the SEM model $F(X)$ takes the following form

$$[\mathrm{F}_\theta(X)]_k = [\mathrm{NN}_\theta(M_{*,k} \odot X)]_k, \tag{13}$$

where $\mathrm{NN}_\theta$ denotes a fully connected neural network function with parameters $\theta$, $M_{*,k}$ denotes the $k$-th column of $\boldsymbol{M}$, and $\odot$ denotes the Hadamard product. The dependency mask is sampled from Gumbel-softmax distribution (Jang et al., 2016), $\boldsymbol{M} \sim p(\boldsymbol{M}|\theta)$ and the parameters $\theta$ are updated during the training (M-step). In this case, the sparsity penalty $\mathcal{R}(\theta)$ in eq. (6) is set as an L1 norm, i.e., $\mathcal{R}(\theta) = \mathbb{E}_{\boldsymbol{M} \sim p(\cdot|\theta)}\|\boldsymbol{M}\|_1$.

**Computing the log-determinant in eq. (12).** We note that $\log \big| \det J_{(\mathrm{id}-\mathbf{DF})}(X)\big| = \log \big| \det(\mathbf{I}-\mathbf{D}J_\mathrm{F})(X)\big|$, where $\mathbf{I}$ is the $K \times K$ identity matrix. Thus, the log-determinant of the Jacobian matrix can be computed using an unbiased estimator based on the power series expansion introduced by Behrmann et al. (2019),

$$\log \big| \det J_{(\mathrm{id}-\mathbf{DF})}(X)\big| = \log \big| \mathbf{I} - J_{\mathbf{DF}}\big| = -\sum_{m=1}^{\infty} \frac{1}{m} \mathrm{Tr}\Big\{J_{\mathbf{DF}}^m(X)\Big\}, \tag{14}$$

where $J_{\mathbf{DF}}^m(X)$ denotes the Jacobian matrix to the $m$-th power. Equation (14) is guaranteed to converge when $\mathrm{F}(X)$ is contractive (Hall, 2013). In practice, eq. (14) is computed by truncating the number of terms in the summation to a finite number. This, however, introduces bias in estimating the log-determinant of the Jacobian. In order to circumvent this issue we follow the steps taken by Chen et al. (2019). The power series expansion is truncated at a random cut-off $N \sim p_\mathbb{N}(N)$, where $p_\mathbb{N}$ is a probability distribution over the set of

---

**Algorithm 1** IMPUTE-REJECTION

---

**Require:** Minibatch data $\mathcal{B} = \{y^{(i)}, r^{(i)}, s^{(i)}\}_{i=1}^{n_B}$, with sampling distribution $q(x)$.
**Ensure:** Imputed data $\tilde{\mathcal{B}} = \{\widehat{x}^{(i)}, r^{(i)}, s^{(i)}\}_{i=1}^{n_B}$.
  1: **for** $i = 1$ to $n_B$ **do**
  2:     Sample $\tilde{x}_{\Omega_i}^{(i)} \sim q(x_{\Omega_i})$.
  3:     Pick $u \sim U[0,1]$.

  4:     **if** $u \leq \frac{p(\tilde{x}_{\Omega_i}^{(i)}, y_{\Gamma_i}^{(i)} | \theta^t) p(r^{(i)} | \tilde{x}_{\Omega_i}^{(i)}, y_{\Gamma_i}^{(i)}, \phi^t)}{c_0 q(\tilde{x}_{\Omega_i}^{(i)})}$ **then**
  5:        Accept sample: $\widehat{x}_{\Omega_i}^{(i)} = \tilde{x}_{\Omega_i}^{(i)}; \quad \widehat{x}_{\Gamma_i}^{(i)} = y_{\Gamma_i}^{(i)}$.
  6:     **end if**
  7: **end for**
       **return** $\tilde{\mathcal{B}} = \{\widehat{x}^{(i)}, r^{(i)}, s^{(i)}\}_{i=1}^{n_B}$

---

natural numbers $\mathbb{N}$. Each term in the finite power series is then re-weighted to obtain the following estimator:

$$\log \big| \det J_{(\mathrm{id}-\mathbf{DF})}(X) \big| = -\mathbb{E}_N \left[ \sum_{m=1}^{N} \frac{\mathrm{Tr}\{J_{\mathbf{DF}}^m(X)\}}{m \cdot P_{\mathbb{N}}(\ell \geq m)} \right] \tag{15}$$

where $P_{\mathbb{N}}$ is the cumulative density function of $p_{\mathbb{N}}$. Gradient calls are still required in the order of $K$. We can reduce this further by using the Hutchinson trace estimator (Hutchinson, 1989), where $W \sim \mathcal{N}(0, \mathbf{I})$:

$$\mathrm{Tr}\left\{ J_{\mathbf{DF}}^m(X) \right\} = \mathbb{E}_W \left[ W^\top J_{\mathbf{DF}}^m(X) W \right], \tag{16}$$

We note that the remainder of eq. (4) can be efficiently computed with a forward pass of the neural network.

### 3.3.2 Calculating Expectation via Imputation

Combining eqs. (4), (8), (15) and (16), we arrive at

$$\mathcal{Q}(\Theta|\Theta^t) \propto \sum_{i=1}^{n} \mathbb{E}_{x_{\Omega_i}^{(i)} | x_{\Gamma_i}^{(i)}, r^{(i)}; \Theta^t} \left\{ \log p\left( r^{(i)} | x_{\Omega_i}^{(i)}, x_{\Gamma_i}^{(i)}, \phi \right) + \log p_{\epsilon_{\mathcal{O}}}\left( \epsilon_{\mathcal{O}_i}^{(i)} \right) - \mathbb{E}_{N,W} \left[ \sum_{m=1}^{N} \frac{W^\top J_{\mathbf{DF}}^m(x^{(i)}) W}{m \cdot P_{\mathbb{N}}(\ell \geq m)} \right] \right\},$$

where $\epsilon^{(i)} = (\mathrm{id} - \mathbf{D}_i \mathrm{F})(x^{(i)})$, $\mathbf{D}_i$ is the diagonal matrix corresponding to the interventional mask for the $i$-th sample, i.e., $\mathbf{D}_i = \mathrm{diag}(s_1^{(i)}, \ldots, s_K^{(i)})$.

The expectation in the approximation for $\mathcal{Q}(\Theta|\Theta^t)$ generally lacks a closed-form solution. Therefore, it must be approximated by the sample mean, using samples drawn from the posterior distribution $p(x_{\Omega_i}^{(i)} | x_{\Gamma_i}^{(i)}, r^{(i)}, \Theta^t)$. This presents two main challenges: (i) the posterior distribution may not have a closed form, and (ii) direct sampling may be infeasible even when the posterior distribution can be evaluated. The difficulty arises due to the presence of nonlinear relations in $\mathrm{F}(X)$ and the missingness mechanism $p(r^{(i)} | x_{\Omega_i}^{(i)}, x_{\Gamma_i}^{(i)}, \phi^t)$, which may preclude straightforward sampling. Therefore, we employ *rejection sampling* (Koller & Friedman, 2009) to draw samples from a proposal distribution $q(x_{\Omega_i}^{(i)})$, from which samples can be readily generated.

To that end, a constant $c_0 > 0$ is chosen such that $c_0 q(x_{\Omega_i}^{(i)}) \geq p(x_{\Omega_i}^{(i)} | x_{\Gamma_i}^{(i)}, r^{(i)}, \Theta^t)$ for all $i = 1, \ldots, n$. However, as stated earlier the posterior distribution is not readily available. Thus, from Bayes rule, we have

$$p(x_{\Omega_i}^{(i)} | x_{\Gamma_i}^{(i)}, r^{(i)}, \Theta^t) = \frac{p(x_{\Omega_i}^{(i)}, x_{\Gamma_i}^{(i)} | \theta^t) p(r^{(i)} | x_{\Omega_i}^{(i)}, x_{\Gamma_i}^{(i)}, \phi^t)}{p(x_{\Gamma_i}^{(i)}, r^{(i)} | \Theta^t)},$$

where the denominator $p(x_{\Gamma_i}^{(i)}, r^{(i)} | \Theta^t)$ can be evaluated using fully observed data. The first term in the numerator can be computed efficiently, as discussed in section 3.3.1. The second term is addressed in section 3.3.3. Before that, we explain how these calculations simplify under more restrictive models. The imputation procedure is summarized in algorithm 1. In our implementation, it is possible that one pass of rejection sampling may not yield enough samples. Thus, we repeat the rejection sampling procedure until we recover at least half of the batch.

### 3.3.3   Missingness Mechanism

In order to compute the log-likelihood in the E-step, we also need to compute $\log p\big(r^{(i)}|x_{\Omega_i}^{(i)}, x_{\Gamma_i}^{(i)}, \phi\big)$, which according to $m$-graphs can be factorized as

$$p(R \mid X, \phi) = \prod_{k=1}^{K} p\big(R_k \mid \mathrm{pa}_{\mathcal{G}_m}(R_k), \phi_k\big). \tag{17}$$

In developing MissNODAG, we focus on a class of MNAR models, where for any $R_k \in R$, $\mathrm{pa}_{\mathcal{G}_m}(R_k) \subseteq X \cup R \setminus X_k, R_k$. According to the results in Nabi et al. (2020), the full law under any MNAR mechanism that follows assumption 1 (i.e, no self-censoring edges and no collider structures) is identified as a function of the observed data distribution. Figure 1 provides two examples of identifiable MNAR mechanisms with $K = 3$ variables.

Under this MNAR class, each conditional factor in the missingness selection mechanism $p(R|X, \phi)$ is modeled using the expit function, with $\phi_k = \{w_k, z_k\}$:

$$p\big(R_k = 0 \mid \mathrm{pa}_{\mathcal{G}_m}(R_k), \phi_k\big) = \mathrm{expit}\big((W_{XR})_{*,k}^{\top}X + (W_{RR})_{*,k}^{\top}R + z_k\big), \tag{18}$$

where $\mathrm{expit}(x) = 1/(1 + e^{-x})$, $\mathbf{W}_{XR}$ and $\mathbf{W}_{RR}$ denote the adjacency matrix corresponding to the edges between $X$ to $R$ and $R$ to $R$ respectively. Maximization during the M-step with respect to $\phi$ reduces to solving a constrained sparsity-regularized logistic regression, with our choice of $\mathcal{R}(\phi) = \sum_{k=1}^{K} \|w_k\|_1$.

Identifiable MNAR mechanisms are particularly well-suited for modeling missingness in cross-sectional studies, surveys, and retrospective analyses. Consider, for instance, a study examining the relationship between smoking $(X_1)$, tar accumulation in the lungs $(X_2)$, and a bronchitis diagnosis $(X_3)$, where missing entries are indicated by $R_1$, $R_2$, and $R_3$, respectively (see figure 1). The edge $X_3 \rightarrow R_1$ reflects that a suspected diagnosis of bronchitis increases the likelihood of inquiring about smoking history. Smokers are more likely to undergo tests for both tar accumulation and bronchitis, represented by $R_2 \leftarrow X_1 \rightarrow R_3$. Moreover, requesting a diagnostic test for bronchitis increases the likelihood of testing for tar, which in turn makes it more likely that smoking status will be queried—captured by $R_1 \leftarrow R_2 \leftarrow R_3$ in figure 1(b); while in the absence of such a dependency, figure 1(a) is most suitable to illustrate the missingness mechanism.

### 3.4   Enforcement of MNAR Identifiability Constraint in the M-step

To ensure that the model class adheres to identifiable MNAR missingness mechanisms, we explicitly incorporate the identifiability conditions (assumption 1) into the optimization objective. The first condition, no self-censoring, can be enforced by masking the diagonal entries of $\mathbf{W}_{XR}$. The second condition, no colliders, requires more care: we must avoid structures of the form $X_k \rightarrow R_j \leftarrow R_k$. Here, the edge $X_k \rightarrow R_j$ corresponds to $(W_{XR})_{k,j}$, and $R_j \leftarrow R_k$ to $(W_{RR})_{k,j}$. To preclude both edges from co-occurring, we require $(W_{XR})_{k,j} \cdot (W_{RR})_{k,j} = 0$ for all $k, j \in [K]$. Define $h_1(\phi) = \|\mathbf{W}_{XR} \odot \mathbf{W}_{RR}\|_1$, where $\odot$ denotes the Hadamard product. Then enforcing $h_1(\phi) = 0$ guarantees the no-collider condition, as formalized below:

**Proposition 3.** *Setting $h_1(\phi) = 0$ is equivalent to requiring $(W_{XR})_{k,j} \cdot (W_{RR})_{k,j} = 0$ for all $k, j \in [K]$.*

*Proof.* This equivalence follows directly by expanding the $\ell_1$-norm in $h_1$; Note that,

$$h_1(\phi) = \|\mathbf{W}_{XR} \odot \mathbf{W}_{RR}\|_1 = \sum_{k,j=1}^{K} |(W_{XR})_{k,j} \cdot (W_{RR})_{k,j}|. \tag{19}$$

Setting $h_1$ to zero in eq. (19) implies that for each $k, j \in [K]$, $(W_{XR})_{k,j} \cdot (W_{RR})_{k,j} = 0$. $\qquad \square$

Beyond the identifiability conditions, we require the missingness indicators to lie outside any cycles—that is, the graph encoding $p(R \mid X)$ must form a directed acyclic graph (DAG). Our parameterization of the missingness mechanism precludes edges of the form $R_j \rightarrow X_i$, so it suffices to constrain $\mathbf{W}_{RR}$ to the space of DAGs. This can be achieved using the trace-exponential acyclicity constraint proposed in Zheng et al. (2018),

namely, $h_2(\phi) = \mathrm{Tr}(e^{\mathbf{W}_{RR} \odot \mathbf{W}_{RR}}) - K = 0$. In summary, during the M-step, the missingness mechanism parameters are updated by solving the following constrained optimization problem:

$$\min_{\phi} \quad -\sum_{i=1}^{n} \mathbb{E}_{x_{\Omega_i}^{(i)} \sim p(\cdot \mid x_{\Gamma_i}^{(i)}, r^{(i)}, \Theta^t)} \left[ \log p\left( r^{(i)} \mid x_{\Gamma_i}^{(i)}, x_{\Omega_i}^{(i)}, \phi \right) \right]$$

$$\text{subject to} \quad \underbrace{\|\mathbf{W}_{XR} \odot \mathbf{W}_{RR}\|_1 = 0}_{h_1(\phi): \text{ no colliders constraint}} , \quad \text{and} \quad \underbrace{\mathrm{Tr}(e^{\mathbf{W}_{RR} \odot \mathbf{W}_{RR}}) - K = 0}_{h_2(\phi): \text{ DAG constraint}}.$$

**Parameter update via Augmented Lagrangian.** Following Zheng et al. (2018), we use the augmented Lagrangian method to solve the constrained optimization problem above. This approach augments the original objective with a quadratic penalty, yielding:

$$\min_{\phi} \quad -\sum_{i=1}^{n} \mathbb{E}_{x_{\Omega_i}^{(i)} \sim p(\cdot \mid x_{\Gamma_i}^{(i)}, r^{(i)}, \Theta^t)} \left[ \log p\left( r^{(i)} \mid x_{\Gamma_i}^{(i)}, x_{\Omega_i}^{(i)}, \phi \right) \right] + \frac{\rho}{2} \sum_{i=1}^{2} h_i(\phi)$$

$$\text{subject to} \quad h_1(\phi) = 0, \quad \text{and} \quad h_2(\phi) = 0,$$

where $\rho > 0$ is the penalty coefficient. A key advantage of the augmented Lagrangian approach is that it approximates the solution to the constrained problem without requiring $\rho$ to tend to infinity. This allows us to reformulate the objective as an unconstrained problem:

$$\min_{\phi} -\sum_{i=1}^{n} \mathbb{E}_{x_{\Omega_i}^{(i)} \sim p(\cdot \mid x_{\Gamma_i}^{(i)}, r^{(i)}, \Theta^t)} \left[ \log p\left( r^{(i)} \mid x_{\Gamma_i}^{(i)}, x_{\Omega_i}^{(i)}, \phi \right) \right] + \frac{\rho}{2} \sum_{i=1}^{2} h_i(\phi) + \sum_{i=1}^{2} \lambda_i h_i(\phi) \tag{20}$$

where $\lambda_1$ and $\lambda_2$ are the dual variables associated with the constraints. These are updated at each iteration via:

$$\lambda_i \leftarrow \lambda_i + \rho h_i(\phi), \quad i = 1, 2. \tag{21}$$

## 4 Experiments

Using both synthetic data and a real-world gene perturbation dataset, we compared MissNODAG[1] to MissDAG (`missdag`) (Gao et al., 2022), an EM-based causal discovery method limited to DAGs and MAR missingness, MVPC (Tu et al., 2019) a constraint-based method capable of handling MNAR missing data. Since MVPC does not handle interventions out-of-the-box, we apply the joint causal inference (JCI) (Mooij et al., 2020) framework along with MVPC to learn from interventional partially observed data. We also tested state-of-the-art imputation methods, including optimal transport (`optransport`) (Muzellec et al., 2020), MissForest (`missforest`) (Stekhoven & Bühlmann, 2012), and mean imputation, followed by causal graph learning from the imputed data using NODAGS-Flow (Sethuraman et al., 2023). Additionally, we also compare MissNODAG with ENCO (Lippe et al., 2022) after imputing the missing data with `optransport` imputation method, see appendix D for details on the implementation of MissNODAG and the baselines. Note that, ENCO, MissDAG, and MVPC are only capable of learning DAGs, putting these methods at a disadvantage compared to the other baselines. Moreover, MissDAG doesn't handle MNAR missingness.

### 4.1 Synthetic Experiments

In all experiments, we generated cyclic directed graphs with K=10 nodes using the Erdős-Rényi (ER) random graph model. For linear SEMs, edge weights were uniformly sampled from $(-0.6, -0.25) \cup (0.25, 0.6)$ and rescaled to ensure contractivity. For nonlinear SEMs, the causal mechanism was defined as $F(X) = \tanh(\mathbf{W}^\top X)$, where the nonzero entries of $\mathbf{W}$ were drawn from the same distribution and similarly rescaled for contractivity. Exogenous noise variables were sampled from a Gaussian distribution with standard deviations randomly chosen between 0.1 and 0.3. The $m$-graph was generated using the ER model, followed by postprocessing to enforce acyclicity in $\mathbf{W}_{RR}$. Missingness mechanism edge weights were sampled from

---

[1]The code for MissNODAG is available at `https://github.com/muralikgs/missnodag`.

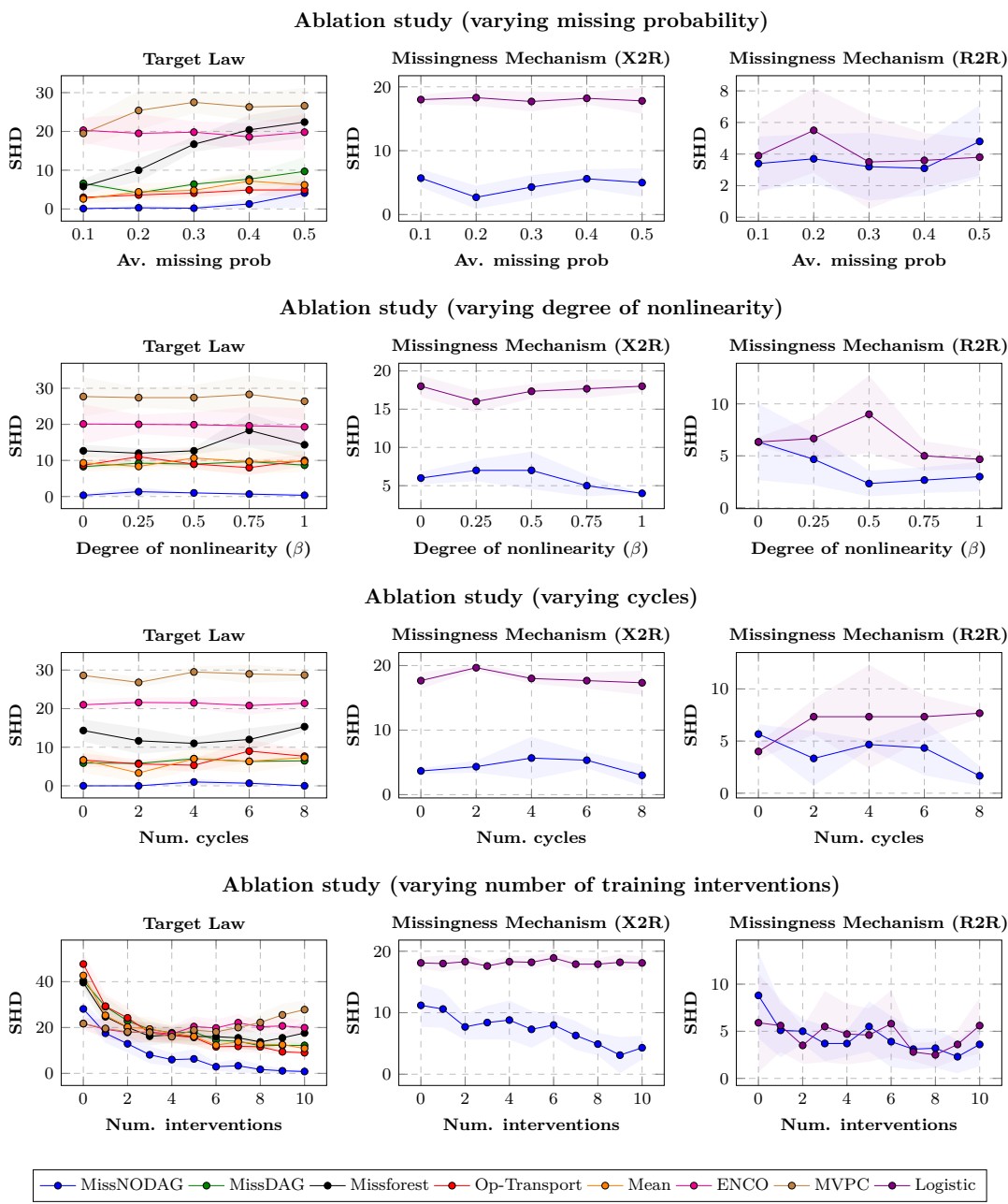

Figure 2: Performance comparison on synthetic data

a standard normal distribution. The training data comprised single-node interventional experiments for all nodes in the target graph (unless specified otherwise). Upon creation of the clean data, some of the observations are replaced with "?" (indicating missing value) based on the missingness mechanism modeled by the $m$-graph. We evaluated MissNODAG and baseline methods across the following five settings.

**Sensitivity to missing probability.** The average probability of a node being missing ($\mathbb{E}[R_i]$) is varied between 0.1 and 0.5. Each interventional setting consists of 1000 samples. Additionally, the outgoing edge densities of the target law and the missingness mechanism are set to 2 and 3 respectively.

Figure 3: Performance comparison between MissNODAG and the baselines as a function of the number of nodes in the target law graph.

**Sensitivity to nonlinearity.**   We varied the degree of nonlinearity by adjusting $\beta \in [0, 1]$ in:

$$X = (1 - \beta)\mathbf{W}^\top X + \beta \tanh(\mathbf{W}^\top X) + \epsilon. \tag{22}$$

Here, $\beta = 0$ corresponds to a fully linear SEM, and $\beta = 1$ to a fully nonlinear SEM. The number of nodes in the target law graph $K$ was set to 10, missing probability set to 0.3, and the number of cycles were randomly chosen.

**Sensitivity to cycles.**   We varied the number of cycles in the target-law graph from 0 to 8, fixing the number of nodes at $K = 10$ and setting $\beta = 1$ in eq. (22). The missing probability was set to 0.3.

**Sensitivity to training interventions.**   We varied the number of interventions provided during the training phase. Starting from 0 interventional experiments in the training set (corresponding to observational setting) to all single-node interventional experiments (10 interventions). Here $K$ was set to 10, missing probability to 0.3, $\beta = 1.0$ in eq. (22).

**Sensitivity to number of nodes.**   In this case, the number of nodes in the target law graph was varied between 10 and 30, with the number of cycles assigned at random and $\beta = 1$ in eq. (22). The missing probability was set to 0.3.

*Structural Hamming distance* (SHD) is used as the error metric to compare MissNODAG with the baselines. SHD counts the number of operations (addition, deletion, and reversal) needed to match the estimated graph to the ground truth.

**Target law recovery.**   We first analyze target law recovery performance, summarized in the left columns of figures 2 and 3. Across all five settings, MissNODAG either outperforms or matches the baselines. In the experiment with varying missingness probabilities (figure 2, top row), the performance of all methods declines slightly as the average missingness probability increases; however, MissNODAG achieves nearly perfect recovery up to a missing probability of 0.3. In the second and third settings (varying nonlinearity and cycle counts), MissNODAG remains robust, achieving SHD scores close to 0. regarding the number of training interventions, MissNODAG performs competitively; while ENCO performs better in the purely observational setting (zero interventions), MissNODAG overtakes it as the number of interventions increases, widening the performance gap. Finally, regarding scalability (figure 3), while all methods deteriorate as the graph size grows, MissNODAG remains the top-performing model for graphs up to $K = 40$ nodes.

**Missingness mechanism recovery.**   For analyzing the missingness mechanism recovery, the baseline model applies logistic regression, treating each missingness indicator $R_i$ as the target and using the remaining indicators $R_{-i}$ along with $X$ as features. We evaluate recovery in two parts: $X \rightarrow R$ edges and $R \rightarrow R$ edges. For the latter, we convert the estimated graph into its completed partially directed acyclic graph

(CPDAG) and compute SHD against the ground truth CPDAG. Results for $X \to R$ edge recovery are shown in the middle column of figures 2 and 3, and for $R \to R$ edge recovery in the right column of figures 2 and 3, with baseline models denoted as `logistic`. On $X \to R$ recovery, MissNODAG consistently outperforms the baselines across all five settings, with a more pronounced performance gap than in target law recovery. For $R \to R$ recovery, MissNODAG again matches or exceeds baseline performance in all settings, though the difference is less pronounced.

## 4.2 Real-World Experiment

Here we present an experiment focused on learning causal graph structure corresponding to a gene regulator network from a single-cell RNA-sequencing (*scRNA-seq*) dataset with genetic interventions. In particular, we focus on the Perturb-CITE-seq dataset (Frangieh et al., 2021), a type of data set that allows one to study causal relations in gene networks at an unprecedented scale. It contains gene expressions taken from 218,331 melanoma cells split into three cell conditions: (i) control (57,627 cells), (ii) co-culture (73,114 cells), and (iii) interferon (INF)-$\gamma$ (87,590 cells).

Due to practical and computational constraints, we restrict our analysis to a subset of 10 genes (out of 20,000), following the experimental setup of Sethuraman et al. (2023), as summarized in table 1. We include all single-gene interventions targeting this subset, treating each cell condition as an independent dataset on which models are trained separately.

A key challenge in scRNA-seq data is the prevalence of zeros, a phenomenon known as *dropout*. While some zeros reflect true biological absence of gene expression (Zappia et al., 2017), others are technical artifacts introduced by the sequencing process (Jiang et al., 2022; Ding et al., 2020). In our work, we treat all zero expression values as missing and apply MissNODAG to impute them during training. Although this approach may misclassify genuine biological zeros as missing, we find that treating zeros as missing improves model performance compared to assuming fully observed data.

Table 1: List of genes chosen from Perturb-CITE-seq dataset (Frangieh et al., 2021).

| STAT1 | B2M | LGALS3 | PTMA |
|-------|-----|--------|------|
| SSR2 | CTPS1 | TM4SF1 | MRPL47 |
| DNMT1 | TMED10 | | |

Since the data set does not provide a ground truth causal graph, it is not possible to directly compare the performance using SHD. Instead, we compare the performance of the causal discovery methods based on its predictive performance over unseen interventions. To that end, we perform a 90-10 split on the three data sets. The smaller set is treated as the test set, which is then used for performance comparison between MissNODAGS and the baselines. As a performance metric, we use the predicted negative log-likelihood (NLL) over the test set after training the models for 100 epochs. The results are summarized in figure 4.

MissNODAG outperforms baseline methods across all three cell conditions. Furthermore, MissNODAG and other imputation-based approaches (`mean` and `optransport`) consistently surpass NODAGS-Flow trained directly on unprocessed data. The middle plot of figure 4 presents a sensitivity analysis in which we vary the proportion of zero gene concentrations treated as missing data (dropouts) from 0% to 100%. The results indicate that predictive performance improves as a larger fraction of zero values is explicitly modeled as dropout. These findings highlight the prevalence of dropouts in scRNA-seq datasets and underscore the critical importance of accounting for them when modeling gene expression data.

Finally, we provide the estimated gene regulatory network in figure 4 (right). MissNODAG has successfully recovered key regulatory structures and feedback loops consistent with established T-cell biology. The directed edge STAT1 $\to$ B2M correctly identifies a primary regulatory mechanism where the transcription factor STAT1 drives the expression of antigen presentation machinery, a well-known response to immune signaling (Frangieh et al., 2021). The hub of connections surrounding CTPS1 validates the model's ability to detect functional metabolic modules, as CTPS1 is the rate-limiting enzyme specifically required for T-cell proliferation and DNA synthesis (Martin et al., 2014). The detected cycle between DNMT1 and STAT1 captures a sophisticated epigenetic feedback loop, where DNMT1 is required for cell division but simultaneously acts to repress STAT1 to prevent excessive inflammation, validating the method's capacity to identify cyclic causal dependencies in biological data (Zhang et al., 2005).

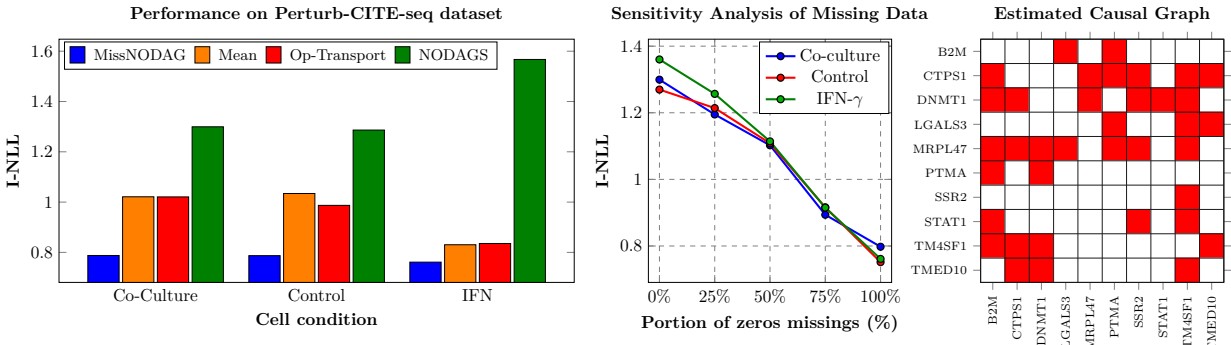

Figure 4: Evaluation on Perturb-CITE-seq gene perturbation data. (Left) Performance comparison on the Perturb-CITE-seq scRNA-seq dataset with $d = 10$ genes, measured using interventional negative log-likelihood (I-NLL). Zero gene expression values are treated as missing. (Middle) Sensitivity analysis for MissNODAG on the different cell conditions with respect to the fraction of zero-valued gene expressions treated as missing; 0% corresponds to the fully observed setting (NODAGS-Flow). (Right) Adjacency matrix of the estimated causal graph for the co-culture cell condition, where red entries indicate the presence of a directed edge from the gene corresponding to the row to the gene corresponding to the column.

## 5 DISCUSSION

In this work, we proposed MissNODAG, a novel differentiable causal discovery framework capable of learning nonlinear cyclic causal graphs along with the missingness mechanism from incomplete interventional data. The framework employs an expectation-maximization algorithm that alternates between imputing missing values and optimizing model parameters. We demonstrated how imputation can be efficiently achieved using rejection sampling, and in the case of linear SEMs with MAR missingness, by directly sampling from the posterior distribution. One of the key strengths of MissNODAG is its ability to handle cyclic directed graphs and MNAR missingness, a significant advancement over methods that typically focus on DAGs and MAR mechanisms.

MissNODAG has three primary limitations. First, our theoretical guarantees rely on strict identifiability assumptions (assumption 1); as shown in appendix C.2, violating these conditions degrades performance, particularly for the recovery of missingness mechanism. Second, the non-convex optimization landscape means the EM procedure guarantees convergence only to a stationary point, not a global maximum. Finally, the log-determinant of the Jacobian computation imposes a computational complexity of $\mathcal{O}(K^3 L)$, limiting scalability to large-scale graphs.

Future research directions include: (1) incorporating realistic measurement noise models in the SEMs to enhance robustness in real-world datasets, as explored in linear DAG models by Saeed et al. (2020); (2) scaling the current framework to larger graphs using low-rank models and variational inference techniques, as explored in acyclic structures by Lopez et al. (2022); (3) allowing for unobserved confounders within the modeling assumptions, as was explored under complete observations by Bhattacharya et al. (2021) and Sethuraman & Fekri (2025); and (4) generalizing our framework to broader classes of identifiable MNAR models, while tackling the challenges of non-identifiability in the full or target laws, as explored in DAG models by Nabi & Bhattacharya (2023) and Guo et al. (2023).

### Acknowledgments

This material is based on work supported National Science Foundation (NSF) under grant number 2502298.

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

# Appendix

The appendix is structured as follows. **Appendix A** offers a summary of the notations used throughout the manuscript for ease of reference. **Appendix B** contains all the proofs. **Appendix C** provides details on three sets of additional simulations and a comparison of training times. **Appendix D** provides additional details on the implementation of the baselines and the MissNODAG framework.

## A GLOSSARY

A comprehensive list of notations used in the manuscript is provided in Table A.1.

Table A.1: Glossary of terms and notations

| Symbol | Definition | Symbol | Definition |
|---|---|---|---|
| $X, x$ | Variables, values | $\mathcal{G}, \mathcal{G}_m$ | Graph, m-graph |
| $\epsilon$ | Exogenous noise terms | $k$ | Graph node index variable |
| $R$ | Missingness indicators | $K$ | Total nodes in $\mathcal{G}$ |
| $Y$ | Coarsened version of $X$ | $\mathrm{pa}_{\mathcal{G}}(X_k)$ | Parent set of $X_k$ in $\mathcal{G}$ |
| $S$ | Intervention indicator vector | $\mathrm{F}, f_k$ | SEM functions |
| $X_{\mathcal{I}}$ | Set of intervened nodes | id | Identity map |
| $X_{\mathcal{O}}$ | Set of non-intervened nodes | $(\mathrm{id} - F)$ | Function mapping $X$ to $\epsilon$ |
| $p_\epsilon(\epsilon)$ | Exogenous noise density function | $J_{\mathrm{F}}(X)$ | Jacobian matrix of F at $X$ |
| $p_X(X), p(X, R)$ | Target, full laws | $\mathbf{I}$ | $K \times K$ Identity matrix |
| $p(Y\|X, R)$ | Coarsening mechanism | $\mathbf{D}$ | Matrix of intervention masks |
| $p(R\|X)$ | Missingness mechanism | $\boldsymbol{B}$ | Weighted adjacency matrix of linear SEM |
| $p(\boldsymbol{M}\|\theta)$ | Gumbel-Softmax distribution for sampling $\boldsymbol{M}$ | $\boldsymbol{M}$ | Dependency mask for SEM function F |
| $\Theta = (\theta, \phi)$ | Parameters of $p_X(X)$, $p(R\|X)$ | $\mathcal{G}_1 \equiv_{\mathcal{I}} \mathcal{G}_2$ | Interventional Markov equivalence |
| $\Theta^t$ | Parameters at EM $t$-th iteration | $\mathbf{D}_i$ | Intervention mask corresponding to $i$-th sample |
| $\Gamma, \Omega$ | Index sets for observed, missing nodes | $i$ | Sample index variable |
| $w_k, z_k$ | Parameters of $p(R_k \mid \mathrm{pa}_{\mathcal{G}_m}(R_k))$ | $n$ | Total sample size |
| $(y^{(i)}, r^{(i)}, s^{(i)})$ | i-th sample | $n_B$ | Mini batch size in each EM |
| $\boldsymbol{A} \odot \boldsymbol{B}$ | Hadamard product of $\boldsymbol{A}$ and $\boldsymbol{B}$ | $m$ | Power series expansion index for log-determinant of Jacobian |
| $\|\cdot\|_1$ | L1 norm | $N$ | Number of power series terms |
| $q(\cdot)$ | Proposal distribution for rejection sampling | $W$ | Gaussian random variable used for computing trace of Jacobian |
| $\mathcal{R}(\cdot)$ | Sparsity inducing regularizer | $e^{\boldsymbol{A}}$ | Matrix exponent of $\boldsymbol{A}$ |
| $h_1(\phi)$ | No colluders constrain function | $h_2(\phi)$ | DAG constraint function |

# B  Theory

## B.1  Joint Density of Target Law Under Intervention

Consider the structural equation model $X = \mathrm{F}(X) + \epsilon$, which implies $X = (\mathrm{id} - \mathrm{F})^{-1}(\epsilon)$. Using the properties of probability density functions, the joint distribution of $X = (X_1, \dots, X_K)$ is given by,

$$p_X(X) = p_\epsilon\big((\mathrm{id} - \mathrm{F})(X)\big)\Big|\det J_{(\mathrm{id}-\mathrm{F})}(X)\Big|, \tag{23}$$

where $p_\epsilon(\epsilon)$ is the probability density function of the exogenous noise vector $\epsilon$. Under an interventional setting $(X_\mathcal{I}, X_\mathcal{O})$, all incoming edges to the nodes in $X_\mathcal{I}$ are removed, leading to the following structural equations:

$$X_k = \begin{cases} \tilde{X}_k & \text{if } X_k \in X_\mathcal{I} \\ f_k\big(\mathrm{pa}_\mathcal{G}(X_k)\big) & \text{if } X_k \notin X_\mathcal{I} \end{cases}$$

That is, $X_k$ is set to a known value $\tilde{X}_k$ if it is intervened upon, and the structural equation remains unchanged when $X_k$ is purely observed. The above equation can be written more concisely as follows:

$$X_k = d_k \cdot \left(f_k(\mathrm{pa}_\mathcal{G}(X_k)) + \epsilon_k\right) + C_k, \quad \text{for } k = 1, \dots, K. \tag{24}$$

where $d_k = \mathbb{1}\{X_k \notin X_\mathcal{I}\}$, and $\mathbb{1}\{\cdot\}$ is the indicator function, and $C_k = \tilde{X}_k$ if $X_k \in X_\mathcal{I}$, and $C_k = 0$ otherwise. Let $\mathbf{D} \in \mathbb{R}^{K \times K}$ be a diagonal matrix such that $D_{kk} = d_k$. Thus, eq. (24) can now be combined for $k = 1, \dots, K$ to obtain the following equation,

$$X = \mathbf{D}\mathrm{F}(X) + \mathbf{D}\epsilon + C, \tag{25}$$

where $F(X)$ is defined in section 2 and $C = (C_1, \dots, C_K)$. Thus we have, $(\mathrm{id} - \mathbf{D}\mathrm{F})(X) = \mathbf{D}\epsilon + C$, this implies that $X = (\mathrm{id} - \mathbf{D}\mathrm{F})^{-1}(\mathbf{D}\epsilon + C)$. Let $X_\mathcal{I} \sim p_{X_\mathcal{I}}(X_\mathcal{I})$. By the change of variable rule in probability, we can write eq. (23) as:

$$p_X(X) = p_{X_\mathcal{I}}(X_\mathcal{I})\, p_{\epsilon_\mathcal{O}}(\epsilon_\mathcal{O})\,\Big|\det J_{(\mathrm{id}-\mathbf{D}\mathrm{F})}(X)\Big|, \tag{26}$$

where $\epsilon_\mathcal{O}$ is the exogenous noise terms of variables in $X_\mathcal{O}$.

## B.2  Convergence Analysis of the EM Algorithm

Here we provide the convergence analysis of MissNODAG. Our analysis relies on the convergence of the EM algorithm (Wu, 1983; Friedman, 1998). The crux of the analysis depends on establishing that the total log-likelihood of the non-missing nodes in the data set either increases or stays the same in each iteration of the algorithm. That is,

$$\sum_{i=1}^{n} \log p\big(x_{\Gamma_i}^{(i)}, r^{(i)} | \Theta^{t+1}\big) \geq \sum_{i=1}^{n} \log p\big(x_{\Gamma_i}^{(i)}, r^{(i)} | \Theta^{t}\big). \tag{27}$$

To that end, note that

$$\sum_{i=1}^{n} \log p(x_{\Gamma_i}^{(i)}, r^{(i)} | \Theta) = \sum_{i=1}^{n} \log p(x_{\Gamma_i}^{(i)} x_{\Omega_i}^{(i)}, r^{(i)} | \Theta) - \sum_{i=1}^{n} \log p(x_{\Omega_i}^{(i)} | x_{\Gamma_i}^{(i)}, r^{(i)}, \Theta).$$

Taking expectation with respect $x_{\Omega_i}^{(i)} | x_{\Gamma_i}^{(i)}, r^{(i)}$ on both side, we get

$$\sum_{i=1}^{n} \log p(x_{\Gamma_i}^{(i)}, r^{(i)} | \Theta) = \sum_{i=1}^{n} \mathbb{E}_{x_{\Omega_i}^{(i)} | x_{\Gamma_i}^{(i)}, r^{(i)}, \Theta^t} \log p(x_{\Gamma_i}^{(i)}, r^{(i)} | \Theta)$$

$$= \underbrace{\sum_{i=1}^{n} \mathbb{E}_{x_{\Omega_i}^{(i)} | x_{\Gamma_i}^{(i)}; \Theta^t} \log p(x_{\Gamma_i}^{(i)} x_{\Omega_i}^{(i)}, r^{(i)} | \Theta)}_{=Q(\Theta | \Theta^t)} - \sum_{i=1}^{n} \mathbb{E}_{x_{\Omega_i}^{(i)} | x_{\Gamma_i}^{(i)}; \Theta^t} \log p(x_{\Omega_i}^{(i)} | x_{\Gamma_i}^{(i)}, r^{(i)}, \Theta). \tag{28}$$

The first term on the RHS in the above equation is nothing but $Q(\Theta|\Theta^t)$. This is maximized in the M-step, i.e., $Q(\Theta|\Theta^{t+1}) \geq Q(\Theta|\Theta^t)$. On the other hand,

$$\sum_{i=1}^{n} \mathbb{E}_{x_{\Omega_i}^{(i)}|x_{\Gamma_i}^{(i)}, r^{(i)}; \Theta^t} \log \frac{p(x_{\Omega_i}^{(i)}|x_{\Gamma_i}^{(i)}, r^{(i)}, \Theta^{t+1})}{p(x_{\Omega_i}^{(i)}|x_{\Gamma_i}^{(i)}, r^{(i)}, \Theta^t)} = -D_{KL}\left(p(x_{\Omega_i}^{(i)}|x_{\Gamma_i}^{(i)}, r^{(i)}, \Theta^t)\|p(x_{\Omega_i}^{(i)}|x_{\Gamma_i}^{(i)}, r^{(i)}, \Theta^{t+1})\right) \leq 0.$$

Thus,

$$\sum_{i=1}^{n} \mathbb{E}_{x_{\Omega_i}^{(i)}|x_{\Gamma_i}^{(i)}, r^{(i)}; \Theta^t} \log p(x_{\Omega_i}^{(i)}|x_{\Gamma_i}^{(i)}, r^{(i)}, \Theta^{t+1}) \leq \sum_{i=1}^{n} \mathbb{E}_{x_{\Omega_i}^{(i)}|x_{\Gamma_i}^{(i)}, r^{(i)}; \Theta^t} \log p(x_{\Omega_i}^{(i)}|x_{\Gamma_i}^{(i)}, r^{(i)}, \Theta^t). \tag{29}$$

From combining eqs. (28) and (29), we can see that at the end of the M-step eq. (27) is satisfied. Similar to the previous results on EM convergence (Wu, 1983; Friedman, 1998), MissNODAG reaches a stationary point of the optimization objective.

## B.3 Consistency of MissNODAG

In this subsection we establish the theoretical foundations for proving consistency of exact maximization of the MissNODAG score function. We begin by reviewing the relevant definition and prior results necessary to prove theorem 2.

### B.3.1 Background

Consider a directed graph $\mathcal{G} = (X, \mathcal{E})$. A *path* $\pi$ between nodes $X_i$ and $X_k$ is a sequence $(X_{i_0}, X_{i_1}, \ldots, X_{i_n})$, with $X_{i_0} = X_i$, $X_{i_n} = X_k$, and every two consecutive nodes in the sequence are connected by an edge, i.e., either $X_{i_j} \to X_{i_{j+1}} \in \mathcal{E}$ or $X_{i_j} \leftarrow X_{i_{j+1}} \in \mathcal{E}$ for all $j = 0, \ldots, n-1$, with $X$. A path is *directed* if all the edges from $X_i$ to $X_k$ are oriented the same way. A cycle through node $X_i$ consists of a directed path from $X_i$ to a node $X_j$ and the directed edge $X_j \to X_i$. For a node $X_i$, the set of *ancestors* is defined as $\text{an}_{\mathcal{G}}(X_i) := \{X_j \in X \mid \text{a directed path exists between } X_j \text{ and } X_i\}$. Similarly, the *descendant* of a node is defined as $\text{de}_{\mathcal{G}}(X_i) := \{X_j \in X \mid \text{a directed path exists between } X_i \text{ and } X_j\}$. The *strongly connected component* of a node $X_i$ is given by the intersection of the ancestors and the descendants of $X_i$, i.e., $\text{sc}_{\mathcal{G}}(X_i) = \text{an}_{\mathcal{G}}(X_i) \cap \text{de}_{\mathcal{G}}(X_i)$. We can apply these definitions to subsets $X_{\mathcal{U}}$ by taking the union over all the elements in the subset, i.e., $\text{de}_{\mathcal{G}}(X_{\mathcal{U}}) = \cup_{i \in \mathcal{U}} \text{de}_{\mathcal{G}}(X_i)$. Finally, a node $X_i$ is called a *collider* in a path $\pi$ if it satisfies the following two conditions: (i) it is a non-endpoint node, and (ii) the subpath $(X_k, X_i, X_j)$ is of the form $X_k \to X_i \leftarrow X_j$.

In an acyclic graph, the notion of *d*-separation is used to relate structural properties in the graph $\mathcal{G}$ to independencies in generated distribution. This is known as the *directed global Markov property* (Forré & Mooij, 2017). However, in general, cyclic graphs do not obey this property and a generalization of *d*-separation in the form of $\sigma$-separation was proposed by Forré & Mooij (2017) to characterize the relationship between graphical properties in a cyclic graph to the independencies observed in the distribution.

**Definition B.1** ($\sigma$-separation). *Let $\mathcal{G} = (X, \mathcal{E})$ be a directe graph and let $X_C \subseteq X$ be a subset of nodes. A path $\pi = (X_{i_0}, X_{i_1}, \ldots, X_{i_n})$ is said to be $\sigma$-blocked given $X_C$ if*

1. *the first node of $\pi$, $X_{i_0} \in X_C$ or its last node $X_{i_n} \in X_C$, or*

2. *$\pi$ contains a collider $X_j \notin X_C$,*

3. *$\pi$ contains a non-collider $X_j \in X_C$ that points towards a neighbor that is not in the same strongly connected component as $X_j$ in $\mathcal{G}$, i.e., $X_k \leftarrow X_j \in \pi$ and $X_k \notin sc_{\mathcal{G}}(X_j)$, or $X_j \to X_\ell \in \pi$ and $X_\ell \notin sc_{\mathcal{G}}(X_j)$.*

*The path $\pi$ is said to be $\sigma$-open given $X_C$ if it is not $\sigma$-blocked. Two subsets of nodes $X_A, X_B \subseteq X$ is said to be $\sigma$-separated given $X_C$ if all the paths between $X_a$ and $X_b$, where $a \in A$ and $b \in B$, are $\sigma$-blocked given $X_C$, and is denoted by*

$$X_A \overset{\sigma}{\underset{\mathcal{G}}{\perp}} X_B \mid X_C.$$

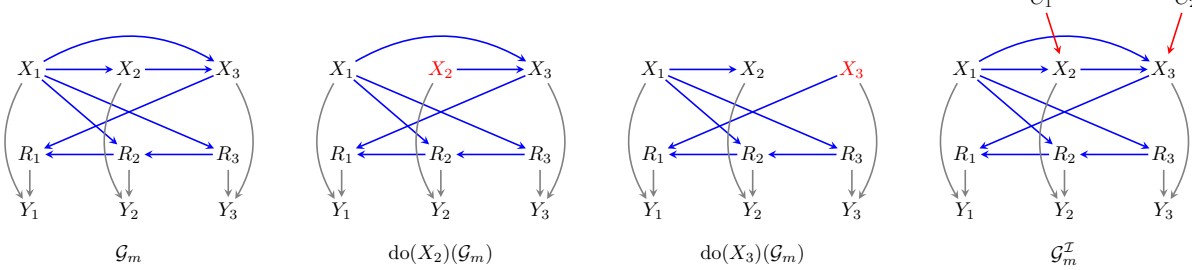

Figure B.1: Illustration of the augmented graph $\mathcal{G}_m^{\mathcal{I}}$ corresponding to the family of interventions $\{X_2\} \cup \{X_3\}$. The graphs $\mathrm{do}(X_2)(\mathcal{G}_m)$ and $\mathrm{do}(X_3)(\mathcal{G}_m)$ corresponding the mutilated graph obtained after hard interventions on $X_2$ and $X_3$ respectively. The augmented graph $\mathcal{G}_m^{\mathcal{I}}$ is the union of the graphs $\mathcal{G}_m$, $\mathrm{do}(X_2)(\mathcal{G}_m)$, and $\mathrm{do}(X_3)(\mathcal{G}_m)$ with the addition of the context variables $C_1$ and $C_2$.

$\sigma$-separation defined above reduces to $d$-separation in acyclic graphs. We can now define the generalized version of directed global Markov property for cyclic graphs.

**Definition B.2** (General directed global Markov property (Forré & Mooij, 2017)). *Let $\mathcal{G} = (X, \mathcal{E})$ be a directed graph and $p$ denote the probability density of the observations $X$. The probability density $p$ satisfies the* general directed global Markov property *if for $X_A, X_B, X_C \subseteq X$*

$$X_A \overset{\sigma}{\underset{\mathcal{G}}{\perp}} X_B \mid X_C \implies X_A \underset{p}{\perp} X_B \mid X_C,$$

*where $X_A \perp_p X_B \mid X_C$, denotes conditionally independence of $X_A$ and $X_B$ given $X_C$ with respect to $p$.*

Note that all the statements made above doesn't exactly depend on the vertex set. Thus, we can replace $\mathcal{G}$ with the m-graph $\mathcal{G}_m$ and retain all the properties stated earlier on the new graph with the vertex set $(X, R)$.

### B.3.2 Joint causal modeling under interventions

To unify multiple interventional settings, we adopt the *joint causal model* Mooij et al. (2020). We create a *meta system* by augmenting the system variables $(X, R)$ with exogenous context variables $C^{\mathcal{I}} = (C_1, \ldots, C_M)$, where each $C_m$ corresponds to a specific intervention target set $X_{\mathcal{I}_m}$. $C_m = \emptyset$ for all $m = 1, \ldots, M$ denotes the observational setting. We construct an augmented graph $\mathcal{G}_m^{\mathcal{I}}$ where the children of each context variable are its targets, that is, $\mathrm{ch}_{\mathcal{G}}(C_m) = X_{\mathcal{I}_m}$ (see figure B.1). Given a family of interventional targets and the corresponding context variables, the structural equations for $X$ take the form:

$$\tilde{f}_k(\mathrm{pa}_{\mathcal{G}_m^{\mathcal{I}}}(X_k), \epsilon_k) = \begin{cases} (C_m)_k, & \text{if } \exists m \in [M] \text{ s.t. } X_k \in I_m, \text{ and } C_m \neq \emptyset, \\ f_k(\mathrm{pa}_{\mathcal{G}}(X_k)) + \epsilon_k, & \text{otherwise.} \end{cases}$$

Note that $\mathrm{pa}_{\mathcal{G}_m^{\mathcal{I}}}(X_k)$ includes nodes from both $X$ and $C^{\mathcal{I}}$. The *context distributions* defined over the context variables $C^{\mathcal{I}}$ do not influence the system in any meaningful way, as noted by Mooij et al. (2020). Moreover, we assume access to the context distribution as the interventional settings are known apriori. For the meta system defined above, the observational distribution corresponds to $p_{\mathcal{G}^{\mathcal{I}}}(X \mid C_1 = \cdots = C_M = \emptyset)$. Similarly, the interventional distribution for the interventional targets $X_{\mathcal{I}_m}$ corresponds to $p_{\mathcal{G}^{\mathcal{I}}}(X \mid C_m = \xi_{\mathcal{I}_m}, C_{-m} = \emptyset)$, i.e.,

$$p_{\mathcal{G}^{\mathcal{I}}}(X \mid C_m = \xi_{\mathcal{I}_m}, C_{-m} = \emptyset) = p_{\mathrm{do}(\mathcal{I}_m)(\mathcal{G}_m)}(X).$$

Here, $p_{\mathrm{do}(\mathcal{I}_m)(\mathcal{G}_m)}$ corresponds to the interventional distribution generated by $\mathcal{G}_m$ for the interventional setting $X_{\mathcal{I}_m}$. Furthermore,

$$p_{\mathcal{G}^{\mathcal{I}}}(C^{\mathcal{I}}, X) = p_{\mathcal{G}^{\mathcal{I}}}(C^{\mathcal{I}}) p_{\mathcal{G}^{\mathcal{I}}}(X \mid C^{\mathcal{I}}). \tag{30}$$

Note that the missingness mechanism is unaltered by augmented the graph with context variables. Thus,

$$p_{\mathcal{G}_m^{\mathcal{I}}}(X, R, C^{\mathcal{I}}) = p_{\mathcal{G}_m^{\mathcal{I}}}(C^{\mathcal{I}}) p_{\mathcal{G}_m^{\mathcal{I}}}(X \mid C^{\mathcal{I}}) p_{\mathcal{G}_m}(R \mid X). \tag{31}$$

Recall, for the interventional setting $X_{\mathcal{I}_m}$, the probability density function governing the observations $X$ is given by eq. (4), which we repeat here for convenience

$$p_{\text{do}(\mathcal{I}_m)(\mathcal{G})}(X) = p_I(X_I)p_\epsilon(\epsilon_\mathcal{O})\big|\det\big(J_{(\text{id}-\mathbf{DF})}(X)\big|,$$

**Definition B.3.** *Let $\mathcal{G} = (X, \mathcal{E})$ be a directed graph, and $\{X_{\mathcal{I}_m}\}_{m=0}^M$ with $X_{\mathcal{I}_0} = X$ be a family of interventional targets. Let $\mathcal{M}_\mathcal{I}(\mathcal{G}_m)$ denote the set of positive densities $p_{\mathcal{G}_m^\mathcal{I}} : \mathbb{R}^{3d} \to \mathbb{R}$ such that $p_{\mathcal{G}_m^\mathcal{I}}$ is given by eq. (31) for all $\mathrm{F} : \mathbb{R}^d \to \mathbb{R}^d$, with $f_i(X) = f_i(pa_\mathcal{G}(X_i))$, such that the resulting forward map $(id - \mathrm{F})$ is invertible, for all $(\sigma_1^2, \ldots, \sigma_K^2)$ such that $\epsilon \sim \mathcal{N}(0, Diag(\sigma_1^2, \ldots, \sigma_K^2))$, and positive density $p_{\mathcal{G}_m^\mathcal{I}}(R \mid X) : \mathbb{R}^{2d} \to \mathbb{R}$.*

**Proposition B.4.** *Let $\mathcal{G} = (X, \mathcal{E})$ be a directed graph, and $\{X_{\mathcal{I}_m}\}_{m=0}^M$ with $X_{\mathcal{I}_0} = X$ be a family of interventional targets, let $p \in \mathcal{M}_\mathcal{I}(\mathcal{G}_m)$, then $p$ satisfies the general directed global Markov property relative to $\mathcal{G}_m^\mathcal{I}$.*

*Proof.* For a directed graph $\mathcal{G}$ and a choice of $\mathrm{F} : \mathbb{R}^d \to \mathbb{R}^d$ such that the forward map $(\text{id} - \mathrm{F})$ is invertible, the structural equations are uniquely solvable with respect to each strongly connected component of $\mathcal{G}$. Moreover, the addition of context variables and the missingness indicators in the augmented graph does not introduce any new cycles. Therefore the meta system forms a simple SCM (Bongers et al., 2021). Thus, from Theorem A.21 in (Bongers et al., 2021), the distribution $p_{\mathcal{G}_m^\mathcal{I}}$ is unique and it satisfies the general *directed global Markov property*. $\square$

We now define the notion of interventional Markov equivalence class for DMGs based on the set of distribution generated by the DMGs.

**Definition B.5** ($\mathcal{I}$-Markov Equivalence Class). *Two directed graphs $\mathcal{G}_m$ and $\mathcal{G}'_m$ are $\mathcal{I}$-Markov equivalent if and only if $\mathcal{M}_\mathcal{I}(\mathcal{G}_m) = \mathcal{M}_\mathcal{I}(\mathcal{G}'_m)$, denoted as $\mathcal{G}_m \equiv_\mathcal{I} \mathcal{G}'_m$. The set of all directed mixed graphs that are $\mathcal{I}$-Markov equivalent to $\mathcal{G}_m$ is the $\mathcal{I}$-Markov equivalence class of $\mathcal{G}_m$, denoted as $\mathcal{I}$-$MEC(\mathcal{G}_m)$.*

### B.3.3 Full law identification from observed law

From the chain rule of probability, the full law $p(X, R)$ is identified if and only if the missingness mechnanism $p(R \mid X)$ is identified. That is,

$$p(X, R) = \frac{p(X, R = 1)}{p(R = 1 \mid X)}p(R \mid X). \tag{32}$$

Nabi et al. (2020) showed that when the $m$-graph satisfies assumption 1, the missingness mechanism is indeed identifiable, with the identifying functional given by

$$p(R \mid X) = \frac{1}{Z} \times \prod_{k=1}^K p(R_k \mid R_{-k}, X) \prod_{k=2}^K \text{OR}(R_k, R_{<k} \mid R_{>k} = 1, X), \tag{33}$$

where $Z$ is the normalization constant, $R_{-k} = R \setminus R_k$, $R_{<k} = \{R_1, \ldots, R_{k-1}\}$, $R_{>k} = \{R_{k+1}, \ldots, R_K\}$, and

$$\text{OR}(R_k, R_{<k} \mid R_{>k} = 1, X) = \frac{p(R_k \mid R_{>k} = 1, R_{<k}, X)}{p(R_k = 1 \mid R_{>k} = 1, R_{<k}, X)} \times \frac{p(R_k = 1 \mid R_{-k} = 1, X)}{p(R_k \mid R_{>k} = 1, R_{<k}, X)}.$$

Note that, under assumption 1, all the terms in the right-hand side of the equation above are functions of the observed distribution, and thus can be estimated from the available data. Furthermore, we make the standard assumption that the missingness mechanism is *strictly positive*. In the following proposition, we establish that the map from observed law to the full law is continuous and injective.

**Proposition B.6.** *Let $\mathcal{G}_m$ be a directed graph satisfying assumption 1. Then, the map $\Psi : p_{obs} \to p_{full}$ given by eq. (32) is injective and continuous.*

*Proof.* Injectivity of the map $\Psi$ is a direct implication of that fact that the missingness mechanism in eq. (33) is identifiable given the observed law, see (Nabi et al., 2020) for more details. Furthermore, continuity of $\Psi$ can be concluded from the fact that the identifying functional in eq. (33) is made up of multiplication and division of positive densities. $\square$

### B.3.4 Proof of Theorem 2

We now provide the proof of the main theoretical result of this paper. Recall the score function introduced in section 3.1,

$$\mathcal{S}(\mathcal{G}_m) := \sup_{\Theta_{\mathcal{G}_m}} \sum_{m=1}^{M} \mathbb{E}_{(X_\Gamma, R) \sim p_{\text{obs}}^{(m)}} \log p_{\text{do}(I_m)(\mathcal{G})}(X_\Gamma, R) - \lambda|\mathcal{G}_m|,$$

where, $p_{\text{obs}}^{(m)}$ is the observed part of the data-generating distribution for $X_{\mathcal{I}_m} \in \{X_{\mathcal{I}_m}\}_{m=0}^{M}$, and $\Theta_{\mathcal{G}_m} = (\theta_{\mathcal{G}_m}, \phi_{\mathcal{G}_m})$ represents the model parameters. In the context of the meta system, the score function above is equivalent to the following score:

$$\mathcal{S}_{\mathcal{I}}(\mathcal{G}_m) := \sup_{\Theta_{\mathcal{G}_m}} \mathbb{E}_{(X_\Gamma, R, C) \sim p_{\mathcal{I}}^*} \log p_{\mathcal{G}_m^{\mathcal{I}}}(X_\Gamma, R, C \mid \Theta_{\mathcal{G}_m}) - \lambda|\mathcal{G}_m|, \tag{34}$$

where, $p_{\mathcal{G}_m^{\mathcal{I}}}(X_\Gamma, R, C \mid \Theta_{\mathcal{G}_m})$ corresponds to the restriction of the complete density function, given by eq. (31), to only the observed nodes. The density $p_{\mathcal{I}}^*$ denotes the joint ground-truth distribution for the observed and the context variables. We define $\mathcal{P}_{\mathcal{I}}(\mathcal{G}_m)$ to be the set of all distributions $p_{\mathcal{G}_m^{\mathcal{I}}}(X, R, C \mid \Theta)$ that can be expressed by the model specified by eqs. (3), (13) and (18). That is,

$$\mathcal{P}_{\mathcal{I}}(\mathcal{G}) := \{p \mid \exists \Theta \text{ s.t } p = p_{\mathcal{G}_m^{\mathcal{I}}}(\cdot \mid \Theta)\}. \tag{35}$$

Note that, $\mathcal{P}_{\mathcal{I}}(\mathcal{G}_m) \subseteq \mathcal{M}_{\mathcal{I}}(\mathcal{G}_m)$. Theorem 2 relies on the following set of assumptions. The first one ensures that the model is capable of representing the ground truth distribution.

**Assumption B.7** (Sufficient Capacity). *The joint ground truth distribution $p_{\mathcal{I}}^*$ is such that $p_{\mathcal{I}}^* \in \mathcal{P}_{\mathcal{I}}(\mathcal{G}_m^*)$, where $\mathcal{G}_m^*$ is the ground truth m-graph.*

In other words, there exists a $\Theta$ such that $p_{\mathcal{I}}^* = p_{\mathcal{G}_m^{\mathcal{I}}}(\cdot \mid \Theta)$. The second assumption is a generalization of the standard faithfulness to the interventional setting for cyclic graphs.

**Assumption B.8** ($\mathcal{I}$-$\sigma$-faithfulness). *Let $V = (X, R, C^{\mathcal{I}})$, for any subset of nodes $A, B, D \subseteq \mathcal{V} \cup C^{\mathcal{I}} \cup R$, and $I_m \in \mathcal{I}$*

$$A \overset{\sigma}{\underset{\mathcal{G}_m^{\mathcal{I}}}{\not\perp}} B \mid B \implies V_A \underset{p_{\mathcal{G}_m^{\mathcal{I}}}}{\not\perp} V_B \mid V_C.$$

As a result of the assumption above, any conditional independency observed in the data must be a consequence of a $\sigma$-separation in the corresponding interventional ground truth graph.

**Assumption B.9** (Finite differential entropy). *For a family of interventional targets $\{X_{\mathcal{I}_m}\}_{m=0}^{M}$,*

$$|\mathbb{E}_{p_{\mathcal{I}}^*} \log p_{\mathcal{I}}^*(X, C)| < \infty.$$

The above assumption ensures that the scenario where $\mathcal{S}(\mathcal{G}^*)$ and $\mathcal{S}(\mathcal{G})$ are both infinity is avoided. This is formalized in the lemma below taken from (Brouillard et al., 2020).

**Lemma B.10** (Finiteness of the score function (Brouillard et al., 2020)). *Under assumptions B.7 and B.9, $|\mathcal{S}_{\mathcal{I}}(\mathcal{G})| < \infty$.*

We finally make the assumption that the density are strictly positive.

**Assumption B.11** (Strict positivity). *For all $V = (X, C, R)$, $p_{\mathcal{G}_m^{\mathcal{I}}}(V) > 0$ for all $p_{\mathcal{G}_m^{\mathcal{I}}} \in \mathcal{M}_{\mathcal{I}}(\mathcal{G}_m)$.*

For the form of score function in eq. (34), Brouillard et al. (2020) showed that the difference in score function between $\mathcal{G}_m^*$ and $\mathcal{G}_m$ can be expressed as the minimization of KL diverengence plus the difference in the regularization terms.

**Lemma B.12** (Rewritting the score function (Brouillard et al., 2020)). *Under assumptions B.7 and B.9, we have*

$$\mathcal{S}(\mathcal{G}_m^*) - \mathcal{S}(\mathcal{G}_m) = \inf_{\Theta} D_{KL}(p_{\mathcal{I},obs}^* \| p_{\mathcal{G}_m^{\mathcal{I}},obs}(\cdot \mid \Theta)) + \lambda(|\mathcal{G}_m| - |\mathcal{G}_m^*|).$$

*where $p_{\cdot,obs}(\cdot)$ denotes the distribution restricted to only the observed variables.*

Furthermore, under complete data, Sethuraman & Fekri (2025) showed that the KL divergence appearing the in lemma above is strictly positive.

**Lemma B.13** (Lemma A.18, (Sethuraman & Fekri, 2025)). *Let $\mathcal{G} = (\mathcal{V}, \mathcal{E})$ be a directed graph, for a set of interventional targets $\{X_{\mathcal{I}_m}\}_{m=0}^M$, and $p^* \notin \mathcal{M}_{\mathcal{I}}(\mathcal{G}_m))$, then*

$$\inf_{p \in \mathcal{M}_{\mathcal{I}}(\mathcal{G}_m))} D(p^* \| p) > 0.$$

We now extend this result to the case of missing data. To that end, we denote $\mathcal{M}_{\mathcal{I},obs}(\mathcal{G}_m)$ to be the set of *observed densities* generated by the graph $\mathcal{G}_m$.

**Lemma B.14.** *Let $\mathcal{G} = (\mathcal{V}, \mathcal{E})$ be a directed graph, for a set of interventional targets $\{X_{\mathcal{I}_m}\}_{m=0}^M$, and $p^*_{obs} \notin \mathcal{M}_{\mathcal{I},obs}(\mathcal{G}_m))$, then*

$$\inf_{p_{obs} \in \mathcal{M}_{\mathcal{I},obs}(\mathcal{G}_m))} D(p^*_{obs} \| p_{obs}) > 0.$$

*Proof.* From lemma B.13, we know that if $p^* \notin \mathcal{M}_{\mathcal{I}}(\mathcal{G}_m))$ then for the full distribution we have

$$\inf_{p \in \mathcal{M}_{\mathcal{I}}(\mathcal{G}_m))} D(p^* \| p) > 0.$$

This implies that $p^*$ does not conform to the independencies enforced by the $\sigma$-separation constraints of $\mathcal{G}_m$. Let us now assume, for the sake of contradiction, that the infimum of the score on the observed data is zero:

$$\inf_{p_{obs} \in \mathcal{M}_{\mathcal{I},obs}(\mathcal{G}_m))} D(p^*_{obs} \| p_{obs}) = 0.$$

By Pinsker's inequality (Cover, 1999), this implies convergence in total variation and thus convergence in distribution. Therefore, there exists a sequence of distributions $\{p_{obs}^{(k)}\}_{k=1}^\infty \subset \mathcal{M}_{\mathcal{I},obs}(\mathcal{G}_m)$ such that:

$$\lim_{k \to \infty} p_{obs}^{(k)} = p^*_{obs}.$$

Let $\Psi$ be the identification mapping in eq. (32) such that $\Psi(p_{obs}) = p$. From proposition B.6, $\Psi$ is injective and continuous. By applying the continuous map $\Psi$ to the sequence we get:

$$\lim_{k \to \infty} \Psi(p_{obs}^{(k)}) = \Psi\left(\lim_{k \to \infty} p_{obs}^{(k)}\right) = \Psi(p^*_{obs}) = p^*.$$

Where the last equality can be obtained by invoking injectivity of the map $\Psi$. Let $g^{(k)} = \Psi(p_{obs}^{(k)})$. Since $p_{obs}^{(k)}$ is generated by the model $\mathcal{G}_m$, its identified full law $g^{(k)}$ must be a valid distribution in the full model $\mathcal{M}_{\mathcal{I}}(\mathcal{G}_m)$.

We now have a sequence $\{g^{(k)}\} \subset \mathcal{M}(\mathcal{G}_m)$ that converges to $p^*$. This would mean that

$$\inf_{p \in \mathcal{M}_{\mathcal{I}}(\mathcal{G}_m))} D(p^* \| p) = 0,$$

which is a contradiction. Thus, we have proved the lemma. $\square$

We are now ready to prove Theorem 2. Recall,

**Theorem 2.** *Let $\{X_{\mathcal{I}_m}\}_{m=0}^M$ be a family of interventional targets, let $\mathcal{G}_m^*$ denote the ground truth directed mixed graph, $p^{(k)}$ denote the data generating distribution for $I_k$, and $\hat{\mathcal{G}}_m := \arg\max_{\mathcal{G}_m \in \mathcal{G}_{id}} \mathcal{S}(\mathcal{G}_m)$. Then, under the Assumptions 1, B.7, B.8, B.9, and B.11, and for a suitably chosen $\lambda > 0$, we have that $\hat{\mathcal{G}}_m \equiv_{\mathcal{I}} \mathcal{G}_m^*$. That is, $\hat{\mathcal{G}}_m$ is $\mathcal{I}$-Markov equivalent to $\mathcal{G}_m^*$.*

*Proof.* The proof follows that of Theorem 2 in Sethuraman & Fekri (2025), with lemma B.14 replacing lemma B.13; we reproduce it here for completeness.

It is sufficient to show that for $\mathcal{G}_m \notin \mathcal{I}\text{-MEC}(\mathcal{G}_m^*)$, the score function of $\hat{\mathcal{G}}_m$ is strictly lower than the score function of $\mathcal{G}_m^*$, i.e., $\mathcal{S}(\mathcal{G}_m^*) > \mathcal{S}(\mathcal{G}_m)$. Since $\mathcal{G}_m \notin \mathcal{I}\text{-MEC}(\mathcal{G}_m^*)$ and $p_\mathcal{I}^* \in \mathcal{M}_\mathcal{I}(\mathcal{G}_m^*)$ (by assumption B.7), there must exist subsets of nodes $A, B, D \subseteq X \cup R \cup C^\mathcal{I}$ such that either:

$$A \overset{\sigma}{\underset{\mathcal{G}_m}{\perp}} B \mid D \quad \text{and} \quad A \overset{\sigma}{\underset{\mathcal{G}_m^*}{\not\perp}} B \mid D, \tag{C1}$$

or

$$A \overset{\sigma}{\underset{\mathcal{G}_m}{\not\perp}} B \mid D \quad \text{and} \quad A \overset{\sigma}{\underset{\mathcal{G}_m^*}{\perp}} B \mid D. \tag{C2}$$

If no such subsets exist, then $\mathcal{G}_m$ and $\mathcal{G}_m^*$ impose the same $\sigma$-separation constraints and thus induce the same set of distributions. This would imply that $\mathcal{G}_m \in \mathcal{I}\text{-MEC}(\mathcal{G}_m^*)$, contradicting our assumption. Since $p_\mathcal{I}^* \in \mathcal{M}_\mathcal{I}(\mathcal{G}_m^*)$, in the case of (C1), it must be true that $A \not\perp_{p_\mathcal{I}^*} B \mid D$ (assumption B.8). Therefore $p_\mathcal{I}^*$ doesn't satisfy the general directed Markov property with respect to $\mathcal{G}_m^\mathcal{I}$ and hence $p_\mathcal{I}^* \notin \mathcal{M}_\mathcal{I}(\mathcal{G}_m)$. For (C2), if $p_\mathcal{I}^* \in \mathcal{M}_\mathcal{I}(\mathcal{G}_m)$, then from assumption B.8, it must be true that $A \not\perp_{p_\mathcal{I}^*} B \mid D$. However, $p_\mathcal{I}^* \in \mathcal{M}_\mathcal{I}(\mathcal{G}_m^*)$, and proposition B.4 implies that $A \perp_{p_\mathcal{I}^*} B \mid D$, resulting in a contradiction. Therefore, $p_\mathcal{I}^* \notin \mathcal{M}_\mathcal{I}(\mathcal{G}_m)$. Moreover, from proposition B.6 we can conclude that $p_{\mathcal{I},obs}^* \notin \mathcal{M}_{\mathcal{I},obs}(\mathcal{G}_m)$.

For convenience, let

$$\eta(\mathcal{G}_m) := \inf_\Theta D_{KL}\big(p_{\mathcal{I},obs}^* \| p_{\mathcal{G}_m^\mathcal{I},obs}(\cdot \mid \Theta)\big).$$

Note that

$$\eta(\mathcal{G}_m) = \inf_\Theta D_{KL}\big(p_{\mathcal{I},obs}^* \| p_{\mathcal{G}_m^\mathcal{I},obs}(\cdot \mid \Theta)\big) \geq \inf_{p \in \mathcal{M}_{\mathcal{I},obs}(\mathcal{G}_m)} D_{KL}(p_{\mathcal{I},obs}^* \| p) > 0,$$

where we use lemma B.14 for the final inequality. Thus, from lemma B.12

$$\mathcal{S}(\mathcal{G}_m^*) - \mathcal{S}(\mathcal{G}_m) = \eta(\mathcal{G}_m) + \lambda(|\mathcal{G}_m| - |\mathcal{G}_m^*|) \tag{36}$$

Following (Brouillard et al., 2020), we now show that by choosing $\lambda$ sufficiently small, the above equation is stictly positive. Note that if $|\mathcal{G}_m| \geq |\mathcal{G}_m^*|$ then $\mathcal{S}(\mathcal{G}_m^*) - \mathcal{S}(\mathcal{G}_m) > 0$. Let $\mathbb{G}^+ := \{\mathcal{G}_m \mid |\mathcal{G}_m| < |\mathcal{G}_m^*|\}$. Choosing $\lambda$ such that $0 < \lambda < \min_{\mathcal{G}_m \in \mathbb{G}^+} \frac{\eta(\mathcal{G}_m)}{|\mathcal{G}_m^*| - |\mathcal{G}_m|}$ we see that:

$$\lambda < \min_{\mathcal{G}_m \in \mathbb{G}^+} \frac{\eta(\mathcal{G}_m)}{|\mathcal{G}_m^*| - |\mathcal{G}_m|} \tag{37}$$

$$\iff \lambda < \frac{\eta(\mathcal{G}_m)}{|\mathcal{G}_m^*| - |\mathcal{G}_m|} \quad \forall \mathcal{G}_m \in \mathbb{G}^+ \tag{38}$$

$$\iff \lambda(|\mathcal{G}_m^*| - |\mathcal{G}_m|) < \eta(\mathcal{G}_m) \quad \forall \mathcal{G}_m \in \mathbb{G}^+ \tag{39}$$

$$\iff 0 < \eta(\mathcal{G}_m) + \lambda(|\mathcal{G}_m| - |\mathcal{G}_m^*|) = \mathcal{S}(\mathcal{G}_m^*) - \mathcal{S}(\mathcal{G}_m) \quad \forall \mathcal{G}_m \in \mathbb{G}^+. \tag{40}$$

Thus, every graph outside of the general directed Markov equivalence class of $(\mathcal{G}_m^*)^\mathcal{I}$ has a strictly lower score. $\qquad\square$

## C  Additional Experiments

### C.1  Sensitivity to Model Parameters

We conducted an sensitivity analysis comparing MissNODAG with baseline models to assess sensitivity to: (a) number of samples per intervention in the training set, and (b) average edge density of the target law graph. In all cases, graphs were generated using the ER random graph model. Training data consisted of single-node interventions over all nodes in the graph. Exogenous noise variables and $m$-graphs were sampled as described in section 4. In both the cases we generate the data from the nonlinear SEM ($\beta = 1$). Results (averaged over 10 trials) are shown in figure C.1.

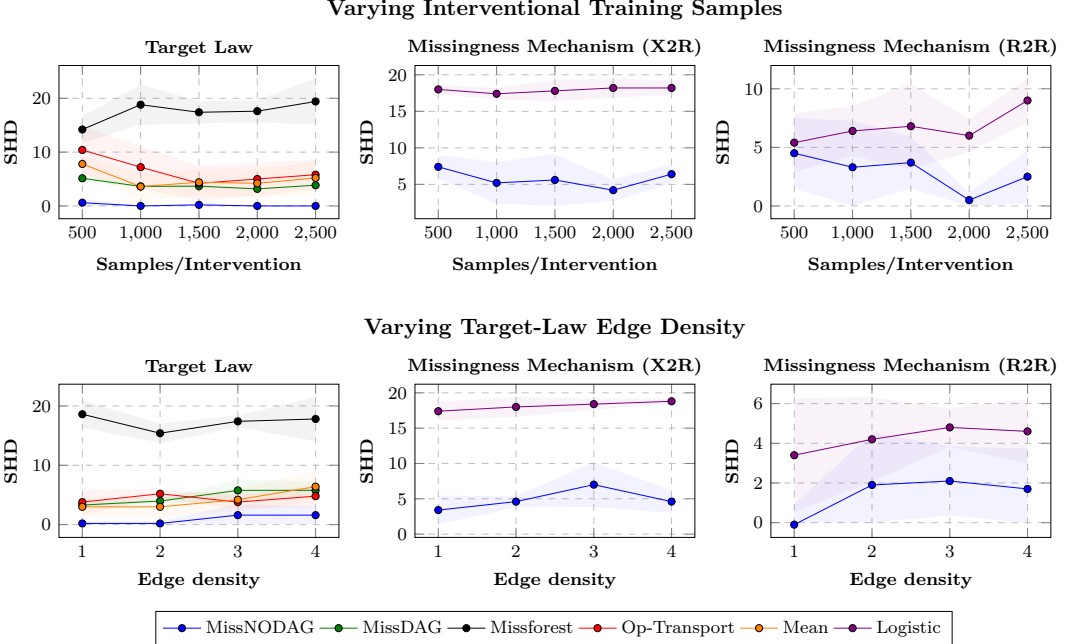

Figure C.1: Performance comparison between MissNODAG and the baselines as a function of: (top row) number samples per intervention in the training data, and (bottom row) edge density of the target-law graph.

**Varying interventional training samples.** With the target law and missingness mechanism outgoing edge densities fixed at 2 and 3, respectively, we varied the number of training samples per interventional setting from 500 to 2,500 (in increments of 500). The average missingness probability was set to 0.3. As seen from figure C.1 (top row), MissNODAG outperforms the baselines in recovering both the target law and the missingness mechanism. MissNODAG achieves near perfect target-law recovery even with 500 samples per intervention. This is showcases the superior sample requirements of MissNODAG compared to the baselines.

**Varying target law edge density.** In our experiments, the outgoing edge density of the target law was varied between 1 and 4, while the average missing probability was fixed at 0.3. Each interventional setting comprised 1,000 samples. From figure C.1 (bottom row), we observe a trend consistent with the previous experiment with MissNODAG outperforming the baselines. However, we observed that the models performed comparably in terms of target law and $R \to R$ edge recovery.

## C.2 Misspecification

We consider three kinds of model misspecification: (i) soft interventions, (ii) non-identifiable missingness mechanism, and (iii) mismatched parametric family for missingness mechanism. In each setting, we considered nonlinear SEM with $K = 10$ nodes, missing probability set to 0.3 and provided all single node interventions as training input.

**Soft interventions.** In this setting, the interventional structural equation model is set as follows:

$$X_k = \begin{cases} \tilde{f}_k\big(\mathrm{pa}_{\mathcal{G}}(X_k)\big) + \tilde{\epsilon}_k & \text{if } X_k \in X_{\mathcal{I}} \\ f_k\big(\mathrm{pa}_{\mathcal{G}}(X_k)\big) + \epsilon_k & \text{if } X_k \notin X_{\mathcal{I}} \end{cases}$$

In our case, we set $\tilde{f}_k = f_k$ and consider $\epsilon_k \sim \mathcal{N}(0, \sigma_k^2)$, $\tilde{\epsilon}_k \sim \mathcal{N}(\mu_k, \sigma_k^2)$, with $\mu_k = 1$ (also known as shift interventions). Thus, the parent-child interactions are preserved under intervention. The results are summarized in figure C.2. As seen from the figure, MissNODAG achieves comparable performance to that of the baselines, albeit achieving higher SHD score compared to hard interventions.

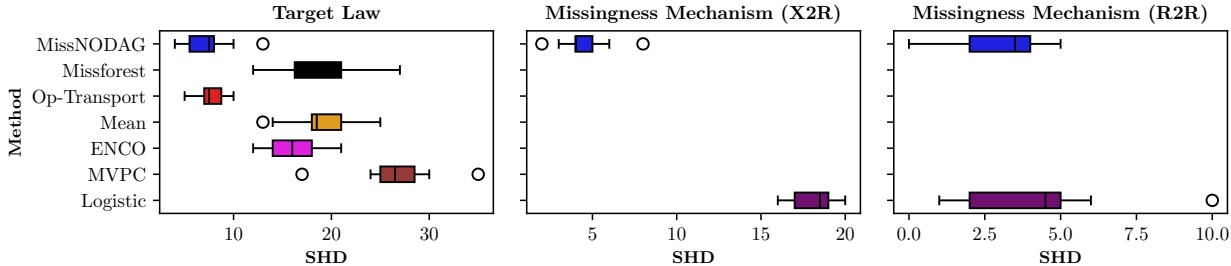

Figure C.2: Performance comparison under soft interventions

**Non-identifiable missingness mechanism.** Here, we explicitly add self-censoring edges as well as colliders into the missingness mechanism while considering hard interventions. The results are summarized in figure C.3. As seen from the figure, MissNODAG is still capable of learning the target law, however, the absolute SHD score increases for missingness mechanism recovery.

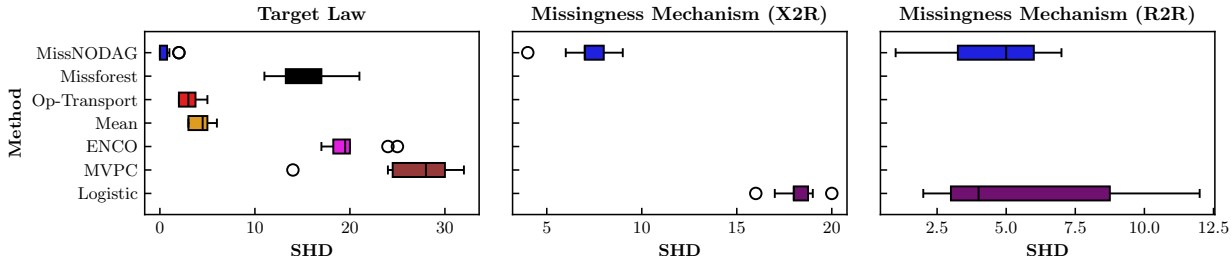

Figure C.3: Performance comparison on non-identifiable missingness mechanism

**Mismatched parametric family for missingness mechanism.** In this setting, we consider the conditional distribution corresponding to each missingness mechanism $R_i$ to be as follows:

$$p\big(R_k = 0 \mid \mathrm{pa}_{\mathcal{G}_m}(R_k), \phi_k\big) = \mathrm{expit}\big(NN_k(X, R) + z_k\big),$$

where $NN_k(X, R)$ is a 2-layer neural network with tanh activation. The results are summarized in figure C.4. As seen from the, the absolute SHD score for target law recovery has increased compared to the well-specified case, however, MissNODAG still retains a comparable performance to that of the baselines.

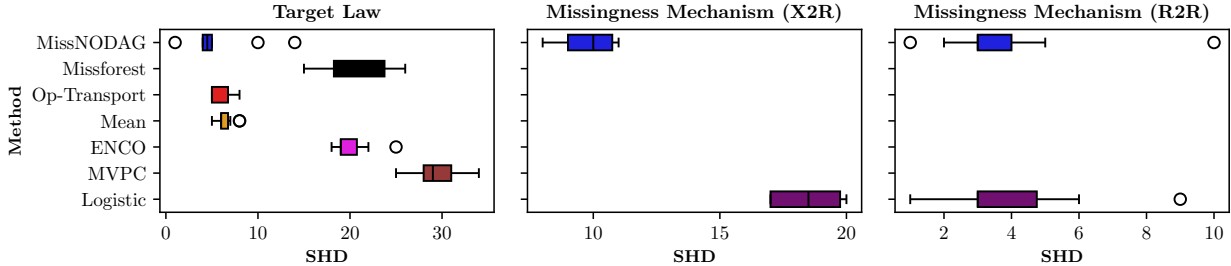

Figure C.4: Performance comparison under missingness mechanism misspecification

## C.3 Computation Time and Convergence Analysis

In figure C.5, we compare the training times of MissNODAG and the baselines for two settings: $K = 10$ nodes (left) and $K = 20$ nodes (right). In both cases, the training data consists of single-node interventions on all

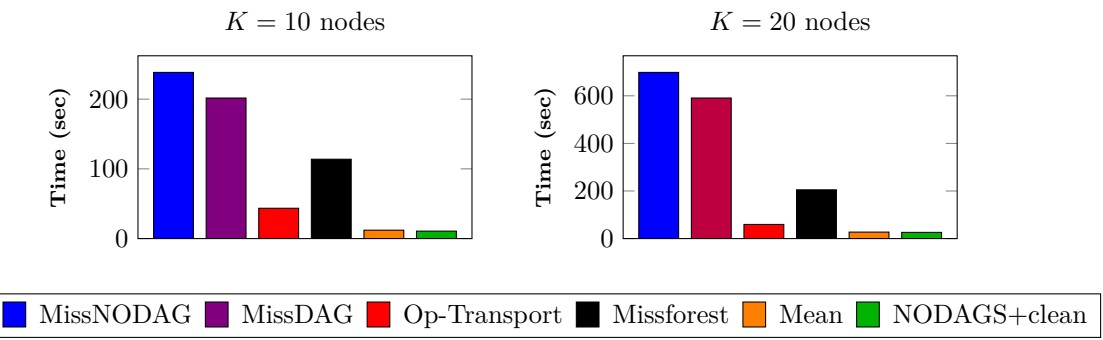

Figure C.5: Training time comparison

nodes, with 500 samples per intervention. MissNODAG and MissDAG were trained for 100 epochs. For other baselines, the reported time includes both data imputation and subsequent training of NODAGS-Flow on the imputed data for 100 epochs. We also report the training time of NODAGS-Flow itself. The two primary computational bottlenecks in MissNODAG are the rejection sampling procedure and the computation of the Jacobian log-determinant, which is required during both the imputation and target law update stages. Utilizing a power series approximation yields a complexity of $\mathcal{O}(K^2)$ for the log-determinant, which can be further reduced in practice via the Hutchinson trace estimator. Consequently, the overall time complexity of MissNODAG is $\mathcal{O}(K^2L)$, where $L$ denotes the maximum number of rejection sampling steps.

As shown in figure C.5, MissNODAG requires the longest training time, though only slightly more than MissDAG in both settings. This additional time is primarily due to jointly learning the missingness mechanism, which MissDAG does not model. As expected, compute time approximately doubles when the number of nodes is doubled.

Additionally, we report the training and eval metrics (log probability of observations, no colluder constraint function value, DAG constraint on $R \rightarrow R$ edges, and target law SHD) as a function training iterations in figure C.6.

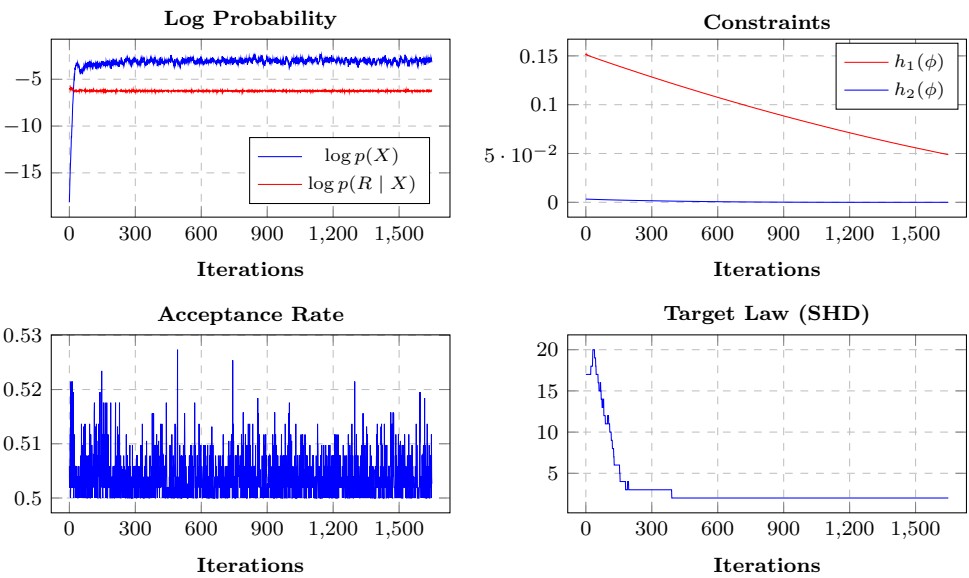

Figure C.6: Training and evaluation metrics as a function of training iterations. Top left: log probability of the observed data for both target law and missingness mechanism. Top right: DAG constraint for $R \rightarrow R$ edges, and no colluder constraint for missingness mechanism. Bottom left: acceptace rate per iteration for rejection sampling. Bottom right: Target law SHD.

### C.4 Sachs Protein Signaling Dataset

We evaluate MissNODAG on a biological dataset for protein signaling network discovery Sachs et al. (2005), a widely used benchmark for causal discovery algorithms.

The dataset contains continuous measurements of multiple phosphorylated proteins and phospholipid components in human immune system cells, with the corresponding network capturing the ordering of interactions among pathway components. Based on $n = 7466$ samples across $m = 11$ cell types, Sachs et al. (2005) identified 20 edges in the underlying graph. Using the consensus network from Sachs et al. (2005) as ground truth, we evaluate performance using the Structural Hamming Distance (SHD) as the error metric. Since the data doesn't contain any missing values, we define a synthetic missingness mechanism to generate missing data from the complete dataset. We set the missing probability to 0.3. The results are summarized in table C.1. We note that the ground-truth graph from (Sachs et al., 2005) is itself a DAG, despite feedback loops being known to exist in these systems (Sturm et al., 2011), as a result, out SHD scores are conservative with respect to MissNODAG's true recovery performance. The recovered directed graph is visualized in figure C.7.

Table C.1: Performance comparison on Sachs et al. (2005) protein signaling dataset.

| Method | TL-SHD |
| --- | --- |
| MissNODAG | **20** |
| MissDAG | 23 |
| Optransport + NODAGS | 22 |
| Optransport + ENCO | 23 |

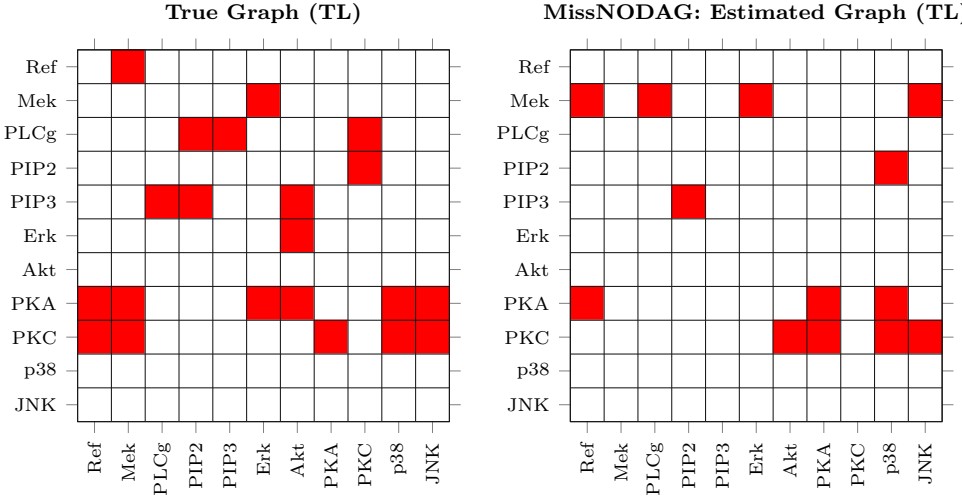

Figure C.7: (Left) Consensus graph from Sachs et al. (2005), (right) graph learned by MissNODAG.

## D Implementation Details

In this section, we describe the implementation details of the MissNODAG framework and the baseline models used for performance comparisons.

**MissNODAG.** We implemented our framework using the Pytorch library in Python and the code used in running the experiments can be found in the `codes` folder within the supplementary materials. We plan to make the code publicly available on GitHub upon publication of the paper.

Starting with an initialization of the model parameters $\Theta^0$, we alternate between the E-step and M-step until the parameters converge. In the E-step, algorithm 1 is used for imputing the missing data, followed by maximizing the expected likelihood of the non-missing nodes in the M-step. We follow the same setup as Sethuraman et al. (2023) for modeling the causal functions, i.e., neural networks (NN) along with dependency mask with entries parameterized by Gumbel-softmax distribution, and for computing the log-determinant of

the Jacobian, i.e., power series expansion followed by Hutchinson trace estimator. Poisson distribution is used for $p_{\mathbb{N}}$ for sampling the number of terms in the expansion to reduce the bias introduced while limiting the number of terms in the power series expansion of log-determinant of the Jacobian, see section 3.3. The final objective in the M-step is maximized using Adam optimizer (Kingma & Ba, 2014).

The learning rate in all our experiments was set to $10^{-2}$. The neural network models used in our experiments contained one multi-layer perceptron layer. No nonlinearities were added to the neural networks for the linear SEM experiments. We used tanh activation for the nonlinear SEM experiments and ReLU activation for the experiments on the perturb-CITE-seq data set. The regularization constant $\lambda$ was set to $10^{-2}$ for the synthetic experiments and $10^{-3}$ for the perturb-CITE-seq experiments. All experiments were performed on NVIDIA RTX6000 GPUs.

**Baselines.** For the baseline NODAGS-Flow[2], we modify the code base provided by Sethuraman et al. (2023) to use the imputed samples for maximizing the likelihood. The hyperparameters of NODAGS-Flow was set to the values described in the previous subsection.

Missforest imputation is performed using the publicly available python library `missingpy`[3]. We use the codebase provided by Muzellec et al. (2020) for optimal transport imputation[4], and the codebase provided by the authors was used for MissDAG[5] (Gao et al., 2022). ENCO (Lippe et al., 2022) was implemented using the codebase[6] provided by the authors with the default hyperparameters. Finally, MVPC was implemented using the `causal-learn`[7] python library (Zheng et al., 2024). The default parameters are used for Missforest, optimal transport imputation, and MissDAG.

---

[2]`https://github.com/Genentech/nodags-flows`
[3]`https://pypi.org/project/missingpy/`
[4]`https://github.com/BorisMuzellec/MissingDataOT`
[5]`https://github.com/ErdunGAO/MissDAG`
[6]`https://github.com/phlippe/ENCO`
[7]`https://github.com/py-why/causal-learn?tab=readme-ov-file`

