# OpenReview forum: "MissNODAG: Differentiable Learning of Cyclic Causal Graphs from Incomplete Data"
_TMLR — Accepted by TMLR_

### Review · Reviewer_DQNF · 2025-11-17

**Summary Of Contributions:**

The paper proposes a unified framework for learning causal graphs with cycles jointly with an MNAR missingness mechanism using a differentiable EM algorithm. It combines contractive flows for the substantive model with a constrained DAG over missingness indicators, achieving identifiability under specific assumptions. The empirical results show improvements over existing approaches on both synthetic and a biological dataset.

**Strengths:** real-world motivated problem, clear formulation, principled identifiability assumptions, well-designed EM procedure, strong synthetic performance.

**Weaknesses:** strong assumptions on interventions and missingness structure, limited real-world stress-testing of MNAR assumptions, and scaling concerns for the E-step.

**Additional Comments:**

Overall, this is a strong and well-presented paper with meaningful contributions. Addressing the above points, especially around assumptions and robustness, would make the work more convincing and broadly applicable.

**Audience:**

Yes

**Audience Explanation:**

Causal discovery with MNAR missingness and cycles is a challenging and active research area; the paper addresses a gap in current methods and provides a principled solution.

**Broader Impact Concerns:**

The method relies on interventional data and strong assumptions on missingness mechanisms; misuse or misinterpretation may lead to incorrect causal conclusions if these assumptions do not hold in real applications (e.g., biology or healthcare). These concerns should be stated more explicitly in the Broader Impact section.

**Claims And Evidence:**

Yes

**Claims Explanation:**

The theoretical assumptions are clearly stated, the EM derivation is sound, and synthetic experiments validate the core claims. The real-world dataset analysis is less conclusive due to simplifying assumptions about missingness but still informative.

**Requested Changes:**

Rather critical:
- Clarify rejection-sampling details and provide diagnostics (acceptance rates, constraint activity).
- Provide experiments with partial or weak interventions to test robustness.
- Add discussion and/or experiments evaluating violations of the “no self-censoring” and “no colluders” assumptions.
- Discuss practical limitations more explicitly.


Non-critical but valuable:
- Report convergence behavior and computational cost.
- Broaden evaluation metrics (e.g., causal effect recovery).
- Release anonymized code for reproducibility (would be valuable for reviewing process)
- Add sensitivity analysis for treating zeros as missing in the Perturb-CITE-seq dataset.

---

> ### Author Response · Authors · 2026-01-06
>
> We thank the reviewer for their thoughtful and constructive feedback. We appreciate the reviewer’s positive assessment of our problem motivation and the design of our EM procedure. Below, we address the specific comments and questions raised.
>
> 	Clarify rejection-sampling details and provide diagnostics (acceptance rates, constraint activity).
>
> We have expanded Subsection 3.3.2 to include additional details regarding the rejection sampling-based imputation procedure. Furthermore, we now provide detailed training diagnostics, including training metrics, evaluation metrics, and acceptance rates, in Figures 12 and 13 of the Appendix.
>
> 	Provide experiments with partial or weak interventions to test robustness. and experiments evaluating violation of missingness mechanism identifiability
>
> We have added experiments in Appendix C.2 comparing MissNODAG against baselines under two types of model misspecification:
> - **Interventional Misspecification:** We evaluated performance under soft interventions (specifically shift interventions), where parent-child relations are retained.
> - **Missingness Mechanism Misspecification:** We tested scenarios involving both violations of missingness mechanism identifiability and misspecification of the parametric conditional distribution.
>
> Additionally, we provide an ablation study on the number of interventions provided during training (see Figure 2, row 4 in Section 4).
>
> 	Discuss practical limitations more explicitly.
>
> We have added an explicit discussion regarding the limitations of our approach in the Discussion section of the paper (Section 5).
>
> 	Report convergence behavior and computational cost.
>
> We report training metrics in Appendix Figure 12, demonstrating the convergence of the SHD score, observed data likelihood, and constraint functions. Regarding computational cost, we discuss scalability with respect to the number of graph nodes. As noted, the primary bottleneck is the Jacobian log-determinant computation, resulting in an overall complexity of $\mathcal{O}(K^2L)$, where $L$ is the maximum number of rejection sampling steps. A training time comparison against baselines is provided in Appendix Figure 11. Finally, we theoretically investigate the convergence of the EM procedure in Appendix B.2.
>
> 	Release anonymized code for reproducibility (would be valuable for reviewing process)
>
> We have included the anonymized as a part of the supplementary files in the submission.
>
> 	Add sensitivity analysis for treating zeros as missing in the Perturb-CITE-seq dataset
>
> We have included a new sensitivity analysis regarding the treatment of zeros as missing data. Results presented in Figure 4 (middle) indicate that performance improves as a higher proportion of zeros are treated as missing data.

---

### Review · Reviewer_FejH · 2025-11-25

**Summary Of Contributions:**

This paper aims to tackle two important problems in causal discovery: (1) handling missing values, and (2) extending the setting to cyclic graphs. To do this, the authors build on MissDAG and use an iterative procedure that interleaves imputation and likelihood maximization to learn both the (possibly cyclic) structure and the missingness mechanism. The evaluation is conducted on synthetic data and real-world single-cell RNA-sequencing data. Overall, I think the paper is technically solid, though some theoretical aspects and parts of the evaluation would benefit from further clarification.

The key strengths are that the paper addresses a genuinely important problem—both missing data and cyclic structures frequently arise in real-world problems and the technical development looks sound; I went through the main components in detail and the methodology seems well-constructed.

As for weaknesses, I would think of two main things: the connections to prior work could be discussed more clearly, and the theoretical identifiability conditions remain a bit vague. Additionally, the evaluation section could be improved (right now is only evaluating on prediction).

**Audience:**

Yes

**Audience Explanation:**

I think both the causal discovery part and the potential application on RNA sequence data will be of interest to the community.

**Claims And Evidence:**

Yes

**Claims Explanation:**

**Strengths**

- I think that although the overall framework is largely built from common modules in causal discovery, such as the used objectives in EM and imputation, as well as contractive residual flows that are widely used for nonlinear additive models, combining these pieces into a single method that works for this specific and realistic setting (cyclic graphs with MNAR missingness) is still non-trivial. I didn’t find any critical issues in how these components are put together.

- The problem setup itself is well motivated, and the synthetic results are convincing to me.

**Weaknesses**

- The most significant one is that the identifiability argument feels unclear or somewhat informal at the moment. I realize this may rely on Sethuraman & Fekri (2025), but I think the paper would benefit from adding a more formal proof or a clearer statement of the identifiability conditions.

- Relatedly, it would be helpful to clarify whether this work (especially the MNAR component)can be viewed as a special case of Sethuraman & Fekri (2025), or how exactly it differs.

- On the real-world evaluation, I understand that there is no ground-truth graph. Still, since this is causal discovery (not just prediction), it would be important to show and discuss the learned structure and check whether at least part of it aligns with existing domain knowledge. I think this would make the evidence that the method is (partially at least) recovering meaningful causal relations.

- A relevant missing baseline  `Lippe, Phillip, Taco Cohen, and Efstratios Gavves, “Efficient neural causal discovery without acyclicity constraints,” ICLR 2022`.

**Requested Changes:**

- A formal identifiability proof

- A clarification on the difference between Sethuraman & Fekri (2025)

- For real-world evaluation, show the learned graph structure and analyze whether it aligns with the domain knowledge

- More baseline comparison as mentioned (though the one I mentioned is not for missing data, you can combine it with imputaion)

---

> ### Author Response · Authors · 2026-01-06
>
> We thank the reviewer for their thoughtful and constructive feedback. We appreciate the reviewer's acknowledgement of the importance of the problem setting and the technical depth of our work. Below, we address the specific points raised.
>
> 	The most significant one is that the identifiability argument feels unclear or somewhat informal at the moment. I realize this may rely on Sethuraman & Fekri (2025), but I think the paper would benefit from adding a more formal proof or a clearer statement of the identifiability conditions.
>
> We thank the reviewer for this crucial observation. We agree that the initial discussion regarding identifiability was informal. In the revised paper, we have provided a rigorous consistency result that extends the theoretical guarantees of Sethuraman & Fekri (2025) [1] to the missing data setting.
>
> The formal theorem is now stated in **Section 3 (Theorem 2)**, with the full proof provided in **Appendix B.3**. The core contribution of this proof relies on **Lemma B.13 (Appendix B.3)**, which establishes the key result necessary to demonstrate consistency. We have also explicitly listed all required assumptions in Appendix B.3.
>
> 	Relatedly, it would be helpful to clarify whether this work (especially the MNAR component)can be viewed as a special case of Sethuraman & Fekri (2025), or how exactly it differs.
>
> Sethuraman & Fekri (2025) [1] address the setting of _unmeasured confounders_, variables that are unobserved across all samples. In contrast, our work addresses the _missing data_ setting, where variables are partially observed (i.e., measured in at least one sample).
>
> Modeling partially observed data using m-graphs and MNAR missingness mechanisms is out of scope for [1]. Therefore, our work should be viewed as a **generalization** of [1] to the missing data regime. While our theoretical framework builds upon the foundation of [1], the extension provided in **Lemma B.13** allows our method to handle the complexities of missing values, making it more broadly applicable.
>
> 	On the real-world evaluation, I understand that there is no ground-truth graph. Still, since this is causal discovery (not just prediction), it would be important to show and discuss the learned structure and check whether at least part of it aligns with existing domain knowledge. I think this would make the evidence that the method is (partially at least) recovering meaningful causal relations.
>
> We agree that grounding the estimated causal graph in reality is essential. While exhaustive experimental verification of every edge is beyond the scope of this work, we provide partial verification of learned edges by cross-referencing them with established domain literature.
>
> MissNODAG successfully recovers key regulatory structures and feedback loops consistent with T-cell biology:
>
> - **STAT1 $\to$ B2M:** This directed edge correctly identifies a primary regulatory mechanism where the transcription factor STAT1 drives the expression of antigen presentation machinery, a well-known response to immune signaling.
>
> - **CTPS1 Hub:** The dense connections surrounding CTPS1 validate the model's ability to detect functional metabolic modules, as CTPS1 is a rate-limiting enzyme specifically required for T-cell proliferation and DNA synthesis.
>
> - **DNMT1 $\leftrightarrow$ STAT1 Cycle:** The detected cycle between DNMT1 and STAT1 captures a sophisticated epigenetic feedback loop; DNMT1 is required for cell division but simultaneously represses STAT1 to prevent excessive inflammation. This validates our method’s capacity to identify cyclic dependencies in real biological data.
>
>
> We have included a discussion of these findings in **Section 4.2**, and the adjacency matrix of the estimated network is provided in **Figure 4**.
>
> 	A relevant missing baseline `Lippe, Phillip, Taco Cohen, and Efstratios Gavves, “Efficient neural causal discovery without acyclicity constraints,” ICLR 2022`.
>
> We thank the reviewer for suggesting this relevant baseline. We have now included **ENCO** in all synthetic experiments presented in the main paper. To adapt ENCO for the missing data setting, we first impute missing values using Optimal Transport (OT) imputation, followed by training ENCO on the imputed data.
>
> [1] Sethuraman, M. G., & Fekri, F. (2025). Differentiable Cyclic Causal Discovery Under Unmeasured Confounders. NeurIPS 2025.

---

> > ### Comment · Reviewer_FejH · 2026-02-04
> >
> > Thanks for the detailed response. Most of my concerns have been addressed with reflection in the revisions.

---

### Review · Reviewer_hP9A · 2025-12-22

**Summary Of Contributions:**

This paper proposes MissNODAG, a differentiable framework for learning cyclic causal graphs from incomplete interventional data under MNAR mechanisms. The work extends EM-based causal discovery to simultaneously handle both cyclic structures and graphically-represented MNAR mechanisms, building upon NODAGS-Flow and MissDAG. The technical approach employs residual normalizing flows with rejection sampling for imputing missing values in cyclic SEMs, and uses augmented Lagrangian methods to enforce identifiable MNAR constraints including no self-censoring and no colluders. The authors provide convergence analysis showing the algorithm reaches a stationary point, and demonstrate superior performance over imputation-then-discovery baselines on synthetic data and gene perturbation datasets.The paper addresses an important problem since real-world systems often contain both feedback loops and missing data, making this a timely and relevant contribution to the causal discovery literature.

**Audience:**

Yes

**Audience Explanation:**

Missing data is important in domains such as biology, healthcare, and economics, where cyclic relationships (e.g., gene regulatory feedback or economic loops) are also common. Most existing methods handle either missingness or cyclicity, but not both. The proposed approach directly addresses this practical gap.

**Broader Impact Concerns:**

I do not see any significant broader impact concerns. The proposed method focuses on methodological advances in causal discovery under missing data and does not raise immediate ethical, safety, or societal risks beyond those common to data-driven modeling approaches.

**Claims And Evidence:**

No

**Claims Explanation:**

While the theoretical framework is sound and the presentation is generally clear, several critical gaps undermine the strength of the evidence.

Q1. The identifiability claims in Section 3.1. The paper states that with complete data, Sethuraman & Fekri (2025) showed the recovered graph belongs to the correct interventional Markov equivalence class, and then suggests similar performance holds under missingness. However, this represents a logical gap. The EM algorithm does not directly optimize the observed data likelihood but rather optimizes a lower bound, which fundamentally changes the optimization landscape. The authors need to either prove that EM stationary points correspond to the correct equivalence class under their missingness assumptions, or provide extensive empirical evidence demonstrating when and why this gap is negligible. Without such analysis, the central claim that MissNODAG recovers the correct causal structure remains inadequately supported.

Q2. The synthetic experiments use only K=10 nodes, which is relatively small and insufficient to demonstrate scalability. A more systematic evaluation over larger graph sizes—analogous to the node-scaling experiments in Section F.5 of MissDAG—would be necessary to assess how performance degrades as problem complexity increases. The ablation study in Appendix C.1 tests up to K=30 and shows performance degradation, but this is not analyzed or discussed. Furthermore, the paper includes only a single real-world dataset from gene perturbation studies, leaving open questions about generalization to other domains where cyclic relationships and MNAR missingness are relevant.

Q3. The baseline comparisons raise concerns about fairness and completeness. In particular, comparing against MissDAG on cyclic graphs is inherently problematic, as MissDAG is explicitly restricted to DAGs. In addition, the experimental evaluation does not include methods specifically designed for MNAR settings. More recent deep learning–based MNAR imputation approaches, followed by graph learning on the imputed data (i.e., impute-then-learn pipelines), are absent.

**Requested Changes:**

C1. Identifiability Analysis (Section 3.1)

The identifiability claims under missing data require further clarification and support. The authors should either
- formally establish that stationary points of the EM objective correspond to the correct interventional Markov equivalence class under the assumed missingness mechanism, or
- provide systematic empirical evidence demonstrating when and why the gap between optimizing a variational lower bound and the observed-data likelihood is negligible in practice.

In addition, the assumptions required for identifiability in the presence of missing data should be stated explicitly and discussed more clearly.

C2. Experiments (Section 4)

- Scalability: Add experiments with larger graphs (e.g., (K=20, 30, 50)) and report computational cost (runtime and memory) as a function of (K). Any observed scalability limitations should be discussed transparently.

- Real-world evaluation: Include at least one additional real-world dataset.

- Baseline comparisons: Improve fairness and coverage of baselines by including MNAR-specific methods or other recent approaches designed for MNAR/cyclic settings. It would also be informative to compare against (a) methods that handle cyclic graphs with complete data, and (b) methods that handle MNAR missingness but assume DAGs. The paper should explicitly acknowledge that comparing MissDAG on cyclic graphs is inherently unfavorable to MissDAG.


C3. Logistic MNAR Assumption

The choice of modeling (p(R \mid X)) using logistic regression, an ablation comparing logistic regression against a small MLP (e.g., a 2-layer network) for (p(R \mid X)) would strengthen the empirical support of the robustness to model misspecification, particularly when the true missingness mechanism deviates from a logistic form.

---

> ### Author Response · Authors · 2026-01-06
>
> We thank the reviewer for their comprehensive summary and constructive feedback. We are encouraged by their recognition of MissNODAG as a 'timely and relevant contribution' to the causal discovery literature, and we appreciate their assessment that our theoretical framework is sound and clearly presented. Below, we address the specific comments and questions raised.
>
> C1. Identifiability Analysis (Section 3.1)
>
>
> 	The identifiability claims under missing data require further clarification and support.
>
> We thank the reviewer for this comment. In response, we have extended our discussion on identifiability to include a formal theorem stating that exact maximization of the regularized observed likelihood recovers a graph within the ground truth Markov equivalence class. This result, detailed in **Theorem 2 (Section 3)**, generalizes prior work on complete data to the missing data setting, demonstrating that the observed likelihood contains sufficient information for recovery (under assumptions detailed in **Appendix B.3**).
>
> Regarding optimization, we acknowledge that EM guarantees convergence only to a stationary point (see **Appendix B.2**) and that using neural networks renders the landscape non-convex, a challenge shared by differentiable causal discovery methods even under complete data. However, our empirical results in **Section 4** demonstrate that this theoretical gap is negligible in practice. Specifically, when the missing probability is low (< 0.3) and single-node interventions are provided, the SHD score for the target law approaches 0. This indicates that the EM procedure effectively recovers the true target law, though learning the missingness mechanism remains more challenging due to the complex constraints required to maintain identifiability.
>
> C2. Experiments (Section 4)
>
> 	Scalability: Add experiments with larger graphs (e.g., (K=20, 30, 50)) and report computational cost (runtime and memory) as a function of (K). Any observed scalability limitations should be discussed transparently.
>
> We thank the reviewer for this comment. We have included an updated ablation study analyzing the effect of graph size on performance, evaluating graphs with $K \in \{10, 20, 30, 40, 50\}$ nodes. As summarized in **Figure 3**, MissNODAG maintains a lower SHD score than all baselines up to $K=40$. In contrast, we observe a sharp spike in error rates for baselines such as ENCO and MVPC, which degrade rapidly as $K$ increases.
>
> Additionally, we provide a computational cost and training time analysis in **Appendix C.3**. As noted, the primary bottleneck is the Jacobian log-determinant computation, resulting in an overall complexity of $\mathcal{O}(K^2L)$, where $L$ is the maximum number of rejection sampling steps. A training time comparison against baselines is provided in **Appendix Figure 11**.
>
> 	Real-world evaluation: Include at least one additional real-world dataset.
>
> We thank the reviewer for their comment. We have included evaluation on another real world dataset. We evaluate MissNODAG on a biological dataset for protein signaling network discovery [1], which is widely used as a benchmark for causal discovery algorithms. The results are presented in Appendix C.4
>
> 	Baseline comparisons: Improve fairness and coverage of baselines by including MNAR-specific methods ...
>
> We thank the reviewer for this comment. We have updated our experiments to include more baselines to cover various aspects of the problem setup. Here we summarize the baselines and they’re capabilities.
>
> | Method               | Cycles | MNAR |
> | -------------------- | ------ | ---- |
> | Missforest + NODAGS  | Yes    | No   |
> | Mean + NODAGS        | Yes    | No   |
> | Optransport + NODAGS | Yes    | Yes  |
> | MissDAG              | No     | No   |
> | JCI + MVPC           | No     | Yes  |
> | Optransport + ENCO   | No     | Yes  |
>
> We have also included explicit statements regarding the inability of MissDAG, ENCO, and MVPC to learn cycles.
>
> C3. Logistic MNAR Assumption
>
> 	The choice of modeling (p(R \mid X)) using logistic regression, an ablation comparing logistic regression ...
> We thank the reviewer for this comment. We have added a separate section in the appendix comparison MissNODAG with the baselines under model misspecification. We considered two types of model misspecification:
> - **Interventional Misspecification:** We evaluated performance under soft interventions (specifically shift interventions), where parent-child relations are retained.
> - **Missingness Mechanism Misspecification:** We tested scenarios involving both violations of missingness mechanism identifiability and misspecification of the parametric conditional distribution. Here, in the second case we consider missingness mechanism relations to be defined by a 2-layer neural network.
>
> The results are reported in Appendix C.2.
>
> [1] Sachs, K., et al. (2005). Causal protein-signaling networks derived from multiparameter single-cell data. Science.

---

### Author Response · Authors · 2026-01-06

We extend our gratitude to the reviewers for their valuable input. In this response, we outline the primary updates to the paper, incorporating their specific recommendations.

### 1. Theoretical Enhancements (Identifiability & Consistency)

- **Formal Identifiability Theorem:** We have added **Theorem 2** in Section 3, providing a formal proof that exact maximization of the regularized observed likelihood recovers a graph within the ground truth Markov equivalence class. This extends the consistency guarantees of Sethuraman & Fekri (2025) to the missing data setting.
- **Proof & Assumptions:** The full proof, including the key **Lemma B.13** and a complete list of necessary assumptions, is now provided in **Appendix B.3**.


### 2. Experimental Expansion (Scalability & Baselines)

- **Scalability Analysis:** We added an ablation study evaluating performance on larger graphs ($K \in \{10, \dots, 50\}$). Results in **Figure 3** show MissNODAG maintains robust performance up to $K=40$, whereas baselines like ENCO and MVPC degrade rapidly.

- **Computational Cost:** We updated **Appendix C.3** to analyze training time and computational complexity (derived as $\mathcal{O}(K^2L)$).

- **New Baselines:** We integrated additional baselines to ensure fair coverage of cyclic and MNAR capabilities, including:

    - **ENCO:** Adapted for missing data via Optimal Transport imputation.

    - **JCI + MVPC:** To cover MNAR settings (assuming acyclicity).

- **Sensitivity Analysis:** We added a sensitivity analysis regarding the treatment of zeros as missing data in the Perturb-CITE-seq dataset (**Figure 4, middle**).


### 3. Robustness & Misspecification

- **Model Misspecification:** We added **Appendix C.2** comparing MissNODAG against baselines under:

    - **Interventional Misspecification:** Using soft (shift) interventions instead of hard interventions.

    - **Missingness Mechanism Misspecification:** Using a 2-layer MLP for the missingness mechanism instead of the assumed logistic regression, and testing violations of identifiability.

- **Ablation on Interventions:** We added an ablation study in **Section 4 (Figure 2, row 4)** evaluating the impact of the number of interventions provided during training.


### 4. Real-World Validation & Diagnostics

- **Biological Validation:** We expanded **Section 4.2** to discuss specific learned edges in the Perturb-CITE-seq dataset that align with established T-cell biology (e.g., STAT1 $\to$ B2M, CTPS1 hubs, and DNMT1 $\leftrightarrow$ STAT1 feedback loops).

- **Sachs Protein signaling data:** Included evaluation on an additional real-world protein signaling dataset.

- **Training Diagnostics:** We added **Figures 12 and 13** in the Appendix, reporting training metrics, evaluation metrics, and rejection sampling acceptance rates.

### 5. Clarifications & Code

- **Limitations:** A new explicit discussion on the practical limitations of the approach was added to **Section 5**.

---

### Decision · Action_Editor_21ji · 2026-03-09

**Recommendation:** Accept as is

**Additional Comments:**

The authors have addressed all reviewers' concerns.

**Audience:**

Yes

**Audience Explanation:**

All reviewers agree that the manuscript has an audience. The paper follows a Missing-Not-At-Random mechanism. The problem setting as well as novel algorithm are definitely relevant to the probabilistic graphical model subcommunity and the TMLR readership. The audience might be relatively smaller as compared to the whole machine learning community, but it is important to let the ML community know of these developments, and to increase the diversity of topics being studied.

**Claims And Evidence:**

Yes

**Claims Explanation:**

All reviewers agree that the manuscript presents appropriate evidence for its claims. The paper studies a (possibly cyclic) structural causal model with additive noise. The authors propose an algorithm to learn the causal graph and the missingness mechanism from partially observed data, by using expectation maximization and imputation. Consistency guarantees are provided. Experimental evidence is provided on synthetic and real world data.